# Convergent evolution links molybdenum insertase domains with organism-specific sequences
Miriam Rabenow, Eduard Haar [ID], Katharina Schmidt, Robert Hänsch, Ralf R. Mendel & Kevin D. Oliphant [ID] ✉

In all domains of life, the biosynthesis of the pterin-based Molybdenum cofactor (Moco) is crucial. Molybdenum (Mo) becomes biologically active by integrating into a unique pyranopterin scaffold, forming Moco. The final two steps of Moco biosynthesis are catalyzed by the two-domain enzyme Mo insertase, linked by gene fusion in higher organisms. Despite well-understood Moco biosynthesis, the evolutionary significance of Mo insertase fusion remains unclear. Here, we present findings from *Neurospora crassa* that shed light on the critical role of Mo insertase fusion in eukaryotes. Substituting the linkage region with sequences from other species resulted in Moco deficiency, and separate expression of domains, as seen in lower organisms, failed to rescue deficient strains. Stepwise truncation and structural modeling revealed a crucial 20-amino acid sequence within the linkage region essential for fungal growth. Our findings highlight the evolutionary importance of gene fusion and specific sequence composition in eukaryotic Mo insertases.

Molybdenum cofactor (Moco) biosynthesis is a ubiquitously present, highly conserved process vital for most eukaryotic organisms and strongly embedded in carbon, nitrogen, and sulfur metabolism[1]. Mutations within the genes encoding Moco biosynthesis lead to a fatal genetic disorder[2]. Human Moco deficiency results in untreatable seizures, intellectual disability, and neonatal death[3].

The transition metal Molybdenum (Mo) is taken up into the cell in the form of molybdate via specific transporters[4,5]. Subsequently, Mo is incorporated into a pyranopterin scaffold to form Moco. In its bound form, Mo can switch between oxidation states IV and VI and catalyze redox reactions involving two electrons[6]. Moco serves as a prosthetic group and is catalytically active in more than fifty enzymes[7]; most Mo enzymes have been found in bacteria[8]. In contrast, five vital Mo enzymes are present in eukaryotes: xanthine dehydrogenase, aldehyde oxidase, sulfite oxidase (SO), nitrate reductase (NR), and mitochondrial amidoxime reducing component[6,9]. Although Moco is present in all plants and mammals, genes encoding Moco biosynthesis or Moco-dependent enzymes could only be found in 45% of fungi[5]. A prominent example of an eukaryotic organism lacking Moco biosynthesis is the model organism *Saccharomyces cerevisiae*[10].

The formation of Moco occurs through a multistep biosynthesis pathway. In eukaryotes, this pathway consists of four steps involving the participation of six proteins[11,12] (Fig. 1a), where the first step occurs in the mitochondria. The cyclic pyranopterin monophosphate (cPMP) is formed

by conversion of guanosine triphosphate[13–15] and exported into the cytoplasm where molybdopterin (MPT) is formed by transferring two sulfur atoms to cPMP[16]. This reaction is catalyzed by the MPT synthase heterotetramer which is re-sulfurized by the MPT synthase sulfurase[13,17].

In *Neurospora crassa*, the final two steps are catalyzed by the Mo insertase NIT-9, which consists of two catalytically active domains connected by a linkage region. Initially, MPT-AMP is formed through adenylation of the phosphate group, catalyzed by the G domain of NIT-9 (NIT-9G)[18]. MPT-AMP is then transferred to the E domain of NIT-9 (NIT-9E), which hydrolyzes MPT-AMP and incorporates Mo into the dithiolene motif, resulting in the prosthetic group Moco[6,19,20]. Recent phylogenetic studies have shown that Mo insertases are highly diverse and can be formed by either two separate proteins or by one protein harboring two domains fused by a versatile linkage region[21].

In the bacterium *Escherichia coli*, the G and E domains are encoded by two separate genes. The two expressed proteins are referred to as MogA (G domain) and MoeA (E domain), and the naming of the domains in other organisms follows the same principle[22]. While in *Homo sapiens* (Gephyrin) and *N. crassa* (NIT-9), the G domain precedes the E domain, the order is reversed in *Arabidopsis thaliana* (CNX1). This suggests that evolutionarily separate events have led to the connection of the two domains by a linkage region. These linkage regions can vary strongly in length and sequence, indicating that Mo insertase has convergently evolved at least twice, once in plants and once in mammals and fungi[23]. Homologies between the

Department of Plant Biology, Technische Universität Braunschweig, Braunschweig, Germany. ✉e-mail: k.oliphant@tu-braunschweig.de

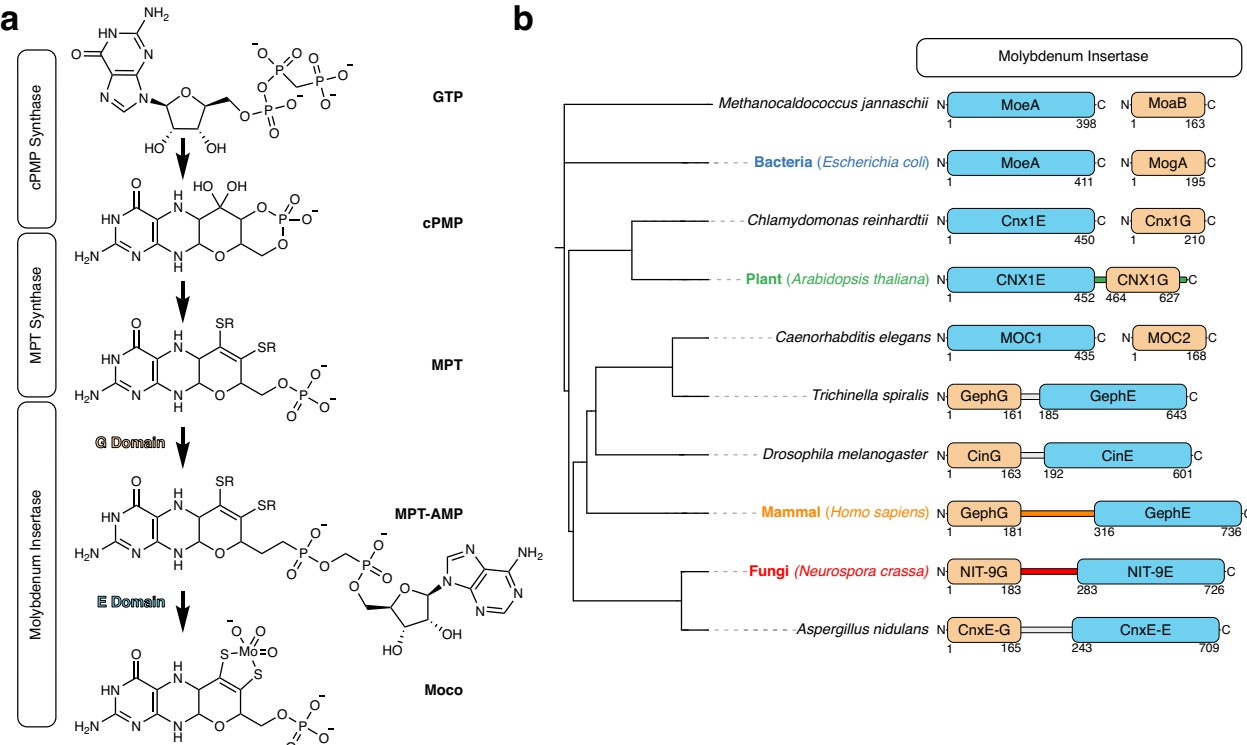

**Fig. 1 | Moco biosynthesis pathway and diversity of the Molybdenum insertase.**
**a** Schematic representation of the Moco biosynthetic pathway with enzyme complexes depicted in boxes. Intermediates such as guanosine triphosphate (GTP), cyclic pyranopterin monophosphate (cPMP), molybdopterin (MPT), molybdopterin adenosine monophosphate (MPT-AMP), and molybdenum cofactor (Moco) structures are illustrated. **b** Phylogenetic tree derived from the NCBI taxonomy database showing selected species for further analysis of the orientation and sequence of the linkage region. Species involved in the analysis are color-coded (Blue: Bacteria; Green: Plant; Orange: Mammal; Red: Fungi). The schematic representation of the Molybdenum insertase protein includes the G domain in beige and the E domain in sky blue, with the linkage region colored according to species. Non-cloned sequences are depicted in gray. Boarders are numbered according to the G or E domain's start and end residue.

insertases can be observed at the functional level. For example, despite having different domain orientations, Gephyrin can restore Moco biosynthesis in Moco-deficient plants, bacteria, and mammals[24]. So far, studies have shown that the two domains of the Mo insertase are fused for higher organisms, whereas for bacteria, archaea, and algae, these genes and proteins are separated[21,25]. In vitro experiments with Gephyrin have previously shown that the linkage region enhances Moco biosynthesis[26]. *Caenorhabditis elegans*, a new model organism for the field of Moco research[27-29], has disrupted this thought and brought up new interest in this subject; it was the first animal discovered with a Mo insertase encoded by two distinct genes. This opened questions such as why the evolutionary union of the Mo insertase is so diverse and if there is a functional benefit to gene fusion.

In this study, we demonstrate that the convergently evolved linkage regions are highly diverse and cannot be interchanged between organisms. To show this, we reversed the orientation of both the domains and linkage region of NIT-9. We introduced linkage sequences from animals, plants, and fungi between the catalytic domains of NIT-9 by creating recombinant strains expressing chimeric proteins. Additionally, we tried to rescue a Δ *nit-9* strain with separately expressed G and E domains, as seen in bacteria. Finally, we stepwise truncated the linkage region, revealing a 20 amino acid sequence that is important for Moco biosynthesis.

## Results
### Mo insertases exhibit high diversity across species
In our endeavor to define Mo insertase sequences more precisely, we compared them mostly across species previously involved in Moco research (Fig. 1b). Sequence alignments were conducted with the full-length fused Mo insertases (Supplementary Fig. 1a), the G domains (Supplementary Fig. 1b), and the E domains (Supplementary Fig. 1c) to identify homologies between species, aiming at elucidating the boundaries of the G and E domains to the linkage region in *N. crassa*. There were no clear boundaries of the G and E domains observable. Therefore, we annotated the protein domains for the species used in this study by utilizing the conserved and previously published residues from *E. coli* (MogA and MoeA), *Methanocaldococcus jannaschii* (MoaB and MoeA), Gephyrin[23,30], and CNX1[20,31] as boundaries.

We conducted a comprehensive phylogenetic analysis of the G and E domain proteins using the Maximum Likelihood method (Supplementary Fig. 2). Phylogenetic trees were constructed based on the JTT matrix-based model, yielding the highest log likelihood values of −9455.98 for the E domain and −4001.28 for the G domain. The phylogeny of the E domain revealed a clade consisting of *A. thaliana* and *Chlamydomonas reinhardtii*, which showed a close relationship to *H. sapiens*. In contrast, the fungal sequences, *N. crassa* and *Aspergillus nidulans*, formed a separate, well-defined clade. The prokaryotic E domains were identified as the most divergent group within this tree. Notably, the *Trichinella spiralis* sequence appeared more closely related to *Drosophila melanogaster* than to its fellow nematode, *C. elegans*. In contrast, the G domain exhibited a different evolutionary pattern, with the *E. coli* sequence clustering with *C. reinhardtii*, adjacent to the fungal clade. These findings indicate a lack of a consistent evolutionary trajectory for the G and E domains, highlighting the intricate complexity of their evolutionary histories.

For eukaryotes, the most significant difference lies in the linkage sequence. In the alga *C. reinhardtii* and the nematode *C. elegans*, the absence of gene fusion separates these proteins[27,32]. This discovery is particularly

interesting because *T. spiralis*, a parasitic nematode and a distant relative to *C. elegans*, possesses a fused Mo insertase. All linked Mo insertases from animals and fungi feature the G domain at the N-terminus and the E domain at the C-terminus. Moco biosynthesis active Gephyrin[33] (Isoform X2) from *H. sapiens* has the longest linkage region among eukaryotes (139 amino acids), which has a functional role in neuroreceptor binding[34]. In contrast, the *N. crassa* linkage region (99 amino acids) constitutes roughly two-thirds of the Gephyrin sequence. Only 10% of the amino acids exhibit sequence similarity between the two examined proteins, compared to 49% sequence similarity between the fungi *A. nidulans* and *N. crassa*. The linkage region of Cinnamon[35] from *D. melanogaster* (28 amino acids) and Gephyrin from *T. spiralis* (24 amino acids) are much shorter than the linkage regions seen in mammals and fungi.

In addition to animal and fungal linkages, plants such as *A. thaliana* developed a different combination by reversing the orientation of the E and G domains, with a short linkage region (11 amino acids)[24,36]. The 163 amino acids long CNX1G is followed by an uncharacterized (43 amino acids) sequence that is not necessary for binding MPT[31]. The CNX1 linkage sequence shows 18% similarity with NIT-9 and 0% with Gephyrin. In contrast, the unspecific region following CNX1G and all linkage regions of fused Mo insertases harbor a conserved histidine residue, which is absent in MogA, MoaB, and MOC2 (Supplementary Fig. 1b).

In summary, our findings highlight the variation of Mo insertases during evolution. Going forward, we will refer to the *N. crassa* linkage sequence as fungal, the Gephyrin linkage sequence as mammal, the CNX1 sequence as plant, and the separate protein as bacterial.

### *N. crassa* cannot grow on nitrate if the Mo insertase linkage region is altered

To understand why Mo insertase genes were repeatedly fused during evolution, we created a homokaryotic (Δ *nit-9*; *his3⁻*) *N. crassa* strain without a Mo insertase and an auxotrophy for histidine (Supplementary Fig. 3). This strain contains an incomplete *his-3* locus, which can be targeted for efficient homologous recombination[37–39].

Subsequently, we generated 18 variations of the Mo insertase, targeting the *his-3* locus, inserting the cassette in reverse orientation, driven by the *ccg1* promoter (*pccg1*)[40]. After transformation, we ensured the strains were homokaryotic by single spore isolation[41] and PCR (Figs. 2a and 3a, Supplementary Fig. 4a). All NIT-9 variants under *pccg1* were expressed more strongly than wild-type NIT-9 (Supplementary Fig. 4b).

The two most vital Moco-dependent enzymes are NR for plants and SO for animals[6]. Both enzymes are present in Moco-utilizing fungi such as *N. crassa*. These fungi can utilize ammonium as a primary nitrogen source, which is Moco-independent, or if primary sources are exhausted, they can switch to secondary nitrogen sources such as nitrate by upregulating their NR machinery[42]. This process is Moco-dependent. Unlike in mammals, depletion of SO activity is not fatal to fungi[13,43]. These traits make fungi unique among eukaryotes, as they can persist without Moco, depending on their nitrogen source. This characteristic allows us to study Moco biosynthesis by observing either NR activity and, thereby, Moco-dependent growth (nitrate) or Moco-independent growth (ammonium). Furthermore, we can utilize chlorate as an NR substrate. If NR activity is present, chlorate is reduced to cytotoxic chlorite, impairing *N. crassa* growth[44].

For the initial *N. crassa* strains, we utilized the fungal, plant, and mammalian linkage regions between the E and G domains from NIT-9 in both orientations (Fig. 2a). For a more detailed examination, we measured the diameters of at least 200 main hyphae from three independent replicates through microscopic analysis (Fig. 2b). We included the wild-type (74-OR-23A; FGSC2489) as well as the knock-out Δ *nit-9* (FGSC18574) strain for comparison in all graphs. On ammonium medium, where all fungi should grow equally, the mean values of hyphal diameters for all strains except for *fun-r* ranged between 6.6–7.3 μm; *fun-r* had an average hyphal diameter of 6.2 μm. Although a change in diameter was observed, no morphological difference could be distinguished between the strains on ammonium medium (Fig. 2c).

The mean diameter of *fun* on nitrate medium (5.2 ± 0.9 μm) was significantly higher than that of *wt* (3.9 ± 0.7 μm). This could be attributed to the NIT-9 overexpression. On nitrate medium, all other variations had significantly smaller diameters than the *wt* and were more comparable to the Δ *nit-9* strain (2.6 ± 0.6 μm). Microscopic images confirmed this, showing thin, highly vacuolized, and stretched cells. *wt* and *fun* could not germinate on chlorate medium due to the produced chlorite. Conversely, the hyphal diameters of the strains expressing the other variants were 4.1–4.7 μm, comparable to Δ *nit-9* (4.3 ± 0.8 μm), although highly vacuolized.

Growth analysis of the *N. crassa* strains was conducted in a race tube assay[45] to support the microscopic results. The overall growth on ammonium, nitrate, and chlorate medium was measured every 24 h for 4 days (Fig. 2d). All strains showed an average total growth of more than 30 cm on ammonium medium, with no significant differences observed compared to *wt*, which was previously reported as 7–8 cm per day[46]. The strain *fun* (34.2 ± 1.9 cm) exhibited similar growth to *wt* (34.3 ± 2.7 cm) on nitrate medium, while they could barely grow on chlorate medium (*wt*: 3.5 ± 1.9 cm; *fun*: 2.4 ± 0.2 cm).

The other strains could keep up with *wt* for the first 24 h on nitrate medium, while on chlorate medium, initial growth was observed after 24 h. After the first day, the strains grown on nitrate medium slowed down, reaching a total growth of 11–19 cm. On chlorate medium, the strains showed almost no signs of chlorite toxification. The results indicated that the overexpression of NIT-9 in *fun* results in thicker hyphae, slightly faster growth on nitrate medium, and a slightly higher sensitivity to chlorate. Overall, the development of *fun-r*, *pla*, *pla-r*, *mam*, and *mam-r* corresponded to the growth progression of Δ *nit-9*.

Additionally, we attempted to recover Δ *nit-9* with the G and E domains from NIT-9 fused by a random, unstructured amino acid sequence from the also linked Moco sulfurase (MOCOS) protein from *H. sapiens* as well as the full-length Gephyrin without success (Supplementary Fig. 5). These results indicate a uniquely developed sequence in the fungal linkage region, which is important for a functional Moco biosynthesis in fungi.

### Separate expression of Mo insertase domains from *N. crassa* cannot rescue Δ *nit-9*

After observing that fusing NIT-9G and NIT-9E with the convergently evolved linkage regions, as well as Gephyrin, could not reconstitute Δ *nit-9*, we investigated whether the bacterial version, where two separate genes encode both domains, could rescue Δ *nit-9*. To determine the importance of gene fusion, we created three strains expressing either NIT-9 without the linkage region, NIT-9G, or NIT-9E (Fig. 3a).

To generate a strain expressing both domains simultaneously, we created a heterokaryon containing nuclei from the *nit-9G* and *nit-9E* strain by mixing conidia at a 1:1 ratio before inoculation[47,48]. Fusion between *N. crassa* cells can occur during germination, leading to the exchange of genetic material[49,50]. As with the previous strains, we measured the diameters of the *w/o* and the *bac* strain through microscopic analysis. No significant deviation of the observed strains from *wt* could be seen on ammonium medium (Fig. 3b). On nitrate medium, the two strains were considerably thinner than *wt* and comparable to Δ *nit-9*. As previously mentioned, *wt* could not germinate on chlorate medium, while *w/o* and *bac* germinated and grew hyphae with diameters between 3.7–4.8 μm.

All variants showed even growth in the race tube assay on ammonium medium (Fig. 3d). For the first 24 h, all strains exhibited the same growth on nitrate, after which *w/o* and *bac* fell behind *wt*, reaching a maximum length of 10–14 cm after 4 days. On chlorate medium, *w/o* and *bac* grew comparable to Δ *nit-9*.

These results indicate that directly connecting the G and E domains without a linkage region reduces fungal growth significantly.

### Mo insertase G domain fused with the linkage region can assemble with the E domain

In addition to the *bac* strain, we created two more heterokaryotic strains to examine the specific function within the linkage region: The G domain in

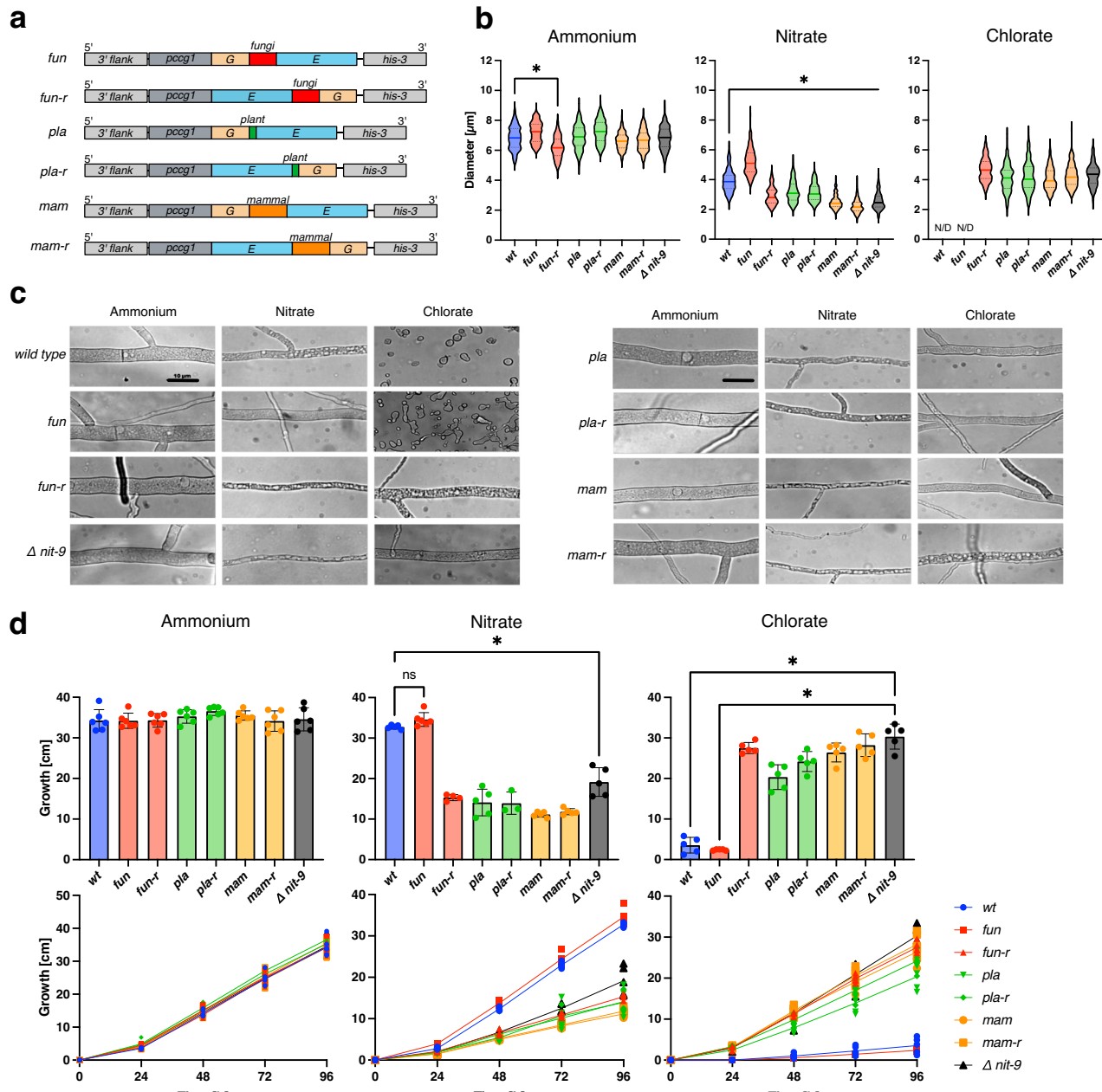

**Fig. 2 | Altering the Mo insertase linkage region from *N. crassa* diminishes growth on nitrate sources. a** Abbreviated schematic of homokaryotic NIT-9 variants expressed by the *ccg1* promoter in the *his-3* locus of an *N. crassa nit-9* knock-out strain. Strain names: *fun* = NIT-9 linked by the 99 amino acid linkage region from fungi. *fun-r* = fungal linkage region in reverse orientation. *pla* = NIT-9 linked by the 11 amino acid linkage region from plant. *pla-r* = plant linkage region in reverse orientation. *mam* = NIT-9 linked by the 139 amino acid linkage region from mammals. *mam-r* = mammalian linkage region in reverse orientation. **b** Diameter of *N. crassa* main hyphae was measured after 24 h of growth on Vogel's medium (MM) agar plates ($n = 200$, mean and 25th–75th percentiles, *$p < 0.001$, one-way ANOVA). N/D non-detectable growth. **c** Hyphae of *N. crassa*, analyzed using bright-field microscopy, bar indicates 10 µm. **d** Race tubes were grown at 30 °C in the darkness; the tubes were marked every 24 h. Growth was measured over 96 h. Experiments were performed in six independent replicates ($n = 3$–6, mean ± SD, *$P < 0.01$, one-way ANOVA). **b–d** All fungi were cultivated with either 80 mM ammonium or 80 mM nitrate as the sole nitrogen source; for samples with 300 mM chlorate, 25 mM ammonium nitrate was used as a nitrogen source. *wt* wild-type strain 74-OR23-1 V. *Δ nit-9 nit-9* knockout strain.

frame with the linkage region (*G-linker*) and the linkage region in frame with the E domain (*linker-E*) (Fig. 3a). Our hypothesis was that the linkage region could enable the reconstitution of the native NIT-9 protein by promoting the assembly of its separate fragments within the cell, as reported for other proteins[51].

Notably, for the microscopic analysis, *G-linker + E* ($3.0 ± 0.6$ µm) had a significantly larger hyphal diameter than *Δ nit-9* ($2.6 ± 0.6$ µm) on nitrate medium (Fig. 3b), although being highly vacuolized (Fig. 3c). On chlorate medium, *G-linker + E* had the thinnest hyphae ($2.6 ± 0.5$ µm) among the

variants. For the race tube assay, *G-linker + E* (14.3 cm) grew better than the second complementation strain *G + linker-E* but did not reach the length of *Δ nit-9* (19.2 cm). On chlorate medium, *G-linker + E* showed the same trend as measured by hyphal diameter, growing 7 cm less than *Δ nit-9*.

Separating the domains, as in bacteria, did not reconstitute *Δ nit-9*. However, connecting the linkage region to the G domain in combination with a separately expressed E domain resulted in a slight recovery of Mo insertase activity, suggesting that not only the linkage but also the specific sequence is important.

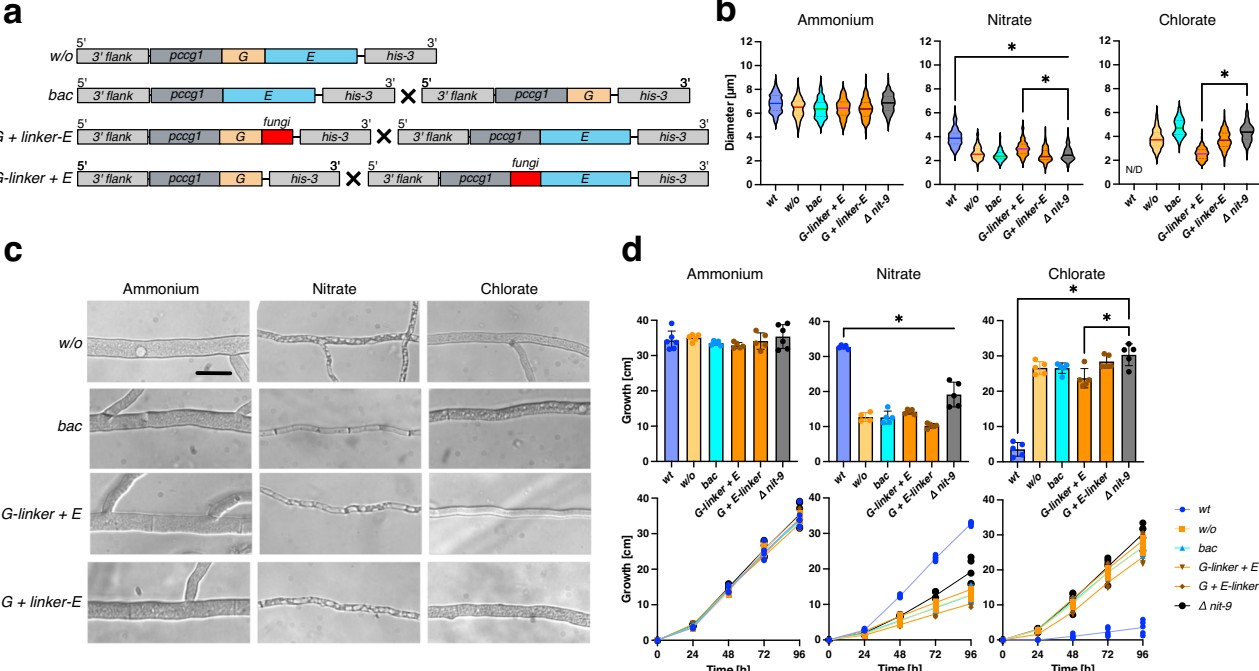

**Fig. 3 | Protein fragment complementation of Mo insertase from *N. crassa* can slightly reconstitute *Δnit-9*. a** Abbreviated schematic of NIT-9 variants expressed by the *ccg1* promoter in the *his-3* locus of an *N. crassa nit-9* knock-out strain. Strains with X were cultivated as heterokaryons. Strain names: *w/o* = NIT-9 domains without linkage region. *bac* = heterokaryotic strain of *nit-9E* and *nit-9G*. *G + linker-E* = heterokaryotic strain of nit-9G and *linker-E*. *G-linker + E* = heterokaryotic strain of *G-linker* and *nit-9E*. **b** Diameter of *N. crassa* main hyphae was measured after 24 h of growth on Vogel's medium (MM) agar plates. (*n* = 200, mean and 25th–75th percentiles, *P < 0.001, one-way ANOVA). **c** Hyphae of *N. crassa*,

analyzed using bright-field microscopy, bar indicates 10 μm. **d** Race tubes were grown at 30 °C in the darkness; the tubes were marked every 24 h. Growth was measured over 96 h. Experiments were performed in six independent replicates (*n* = 4–6, mean ± SD, *P < 0.01, one-way ANOVA). **b–d** All fungi were cultivated with either 80 mM ammonium or 80 mM nitrate as the sole nitrogen source; for samples with 300 mM chlorate, 25 mM ammonium nitrate was used as a nitrogen source. *wt* wild-type strain 74-OR23-1 V. *Δ nit-9 nit-9* knock-out strain. For *bac*, *G-linker + E*, and *G + linker-E*, the conidia were mixed 1:1 before cultivation.

## Mo insertase linkage region truncation unveils a crucial 20-amino acid region important for nitrate-dependent growth

To understand which part of the Mo insertase linkage region is crucial for fungal growth, we truncated the linkage region through a step-by-step deletion of 20 amino acids within the sequence. For this purpose, we created five strains (Fig. 4a), which were analyzed similarly to the previous strains.

No difference between the strains was observed on ammonium medium. However, only *A184-G203* (2.4 ± 0.7 μm) hyphae showed deviations from *wt* on nitrate medium (Fig. 4b). The only strain with a truncation in the NIT-9 linkage region that germinated on chlorate medium was *A184-G203*. The four other truncations showed no significant loss of Mo insertase activity. The hyphae of *A184-G203* and the *Δ nit-9* strain grew similarly, stretched, and loaded with vacuolar structures when grown on nitrate medium (Fig. 4c). *G204-G223*, E224-G243, H244-P263, and *K264-P282* showed no significant difference compared to *wt* on all media. *A184-G203* grew slower than *Δ nit-9* on nitrate medium, although on chlorate medium *Δ nit-9* grew non-significantly faster than *A184-G203* (Fig. 4d), suggesting that there is little NR activity in the *A184-G203* strain.

These results indicate that the amino acids A184-G203 in the NIT-9 linkage region are crucial for proper Mo insertase function and, consequently, for fungal growth on nitrate medium. This specific region appears to play a significant role in the stability or function of the Mo insertase enzyme in *N. crassa*, affecting its ability to utilize nitrate effectively.

## Moco biosynthesis in *N. crassa* relies on NIT-9 A184-G203

We confirmed NR activity within the *A184-G203* strain by switching from in vivo measurements to an in vitro NR assay to detect Moco indirectly[52]. If the conversion from nitrate to nitrite is successful, it can be implicated that Moco biosynthesis is intact. The presence of nitrate as a

nutrient in the growth medium triggers the induction of NR. Therefore, *N. crassa* strains were first cultured in liquid medium containing ammonium to increase mycelial mass. Subsequently, ammonium was washed out of the mycelium and transferred to nitrate medium for 3 h to induce the expression of NR. NR activity was detected through the formation of nitrite[53] (Fig. 5a).

For the strains used during the growth experiments of different linkage variants and orientation as well as the separate expression of the domains no NR activity was measurable, only *wt* showed NR activity (Supplementary Fig. 6). The NR activity of the strains *G204-G223*, *E224-G243*, and *H244-P263* was not significantly higher than that of *wt*. However, for *A184*-G203, there was a fiveold loss in NR activity compared to *wt*, while *Δ nit-9* showed no observable activity. This indicates that *A184-G203* reduces the amount of Moco produced but does not abolish Moco.

In addition, we attempted to analyze the MPT-AMP and Moco/MPT content in these strains by controlled oxidation to the fluorescent derivative FormA[27] to further elucidate the specific stage of Moco biosynthesis disrupted in the A184-G203 strain. All strains tested did not yield quantifiable amounts of either compound, suggesting that the levels may be below the detection limit in our crude extract assays (Supplementary Fig. 7).

## Structural Insights into NIT-9 domains hint at the crucial role of A184-G203

To further investigate the created constructs, we analyzed the protein structures of the convergently evolved linkage regions in the NIT-9 environment using AlphaFold 3[54]. The models revealed that the linkage region in NIT-9 can be divided into a structured and unstructured area (Fig. 5b), with the structural part aligning with A184-G203. To understand where Moco and its intermediates bind within the G and E domains, we superimposed

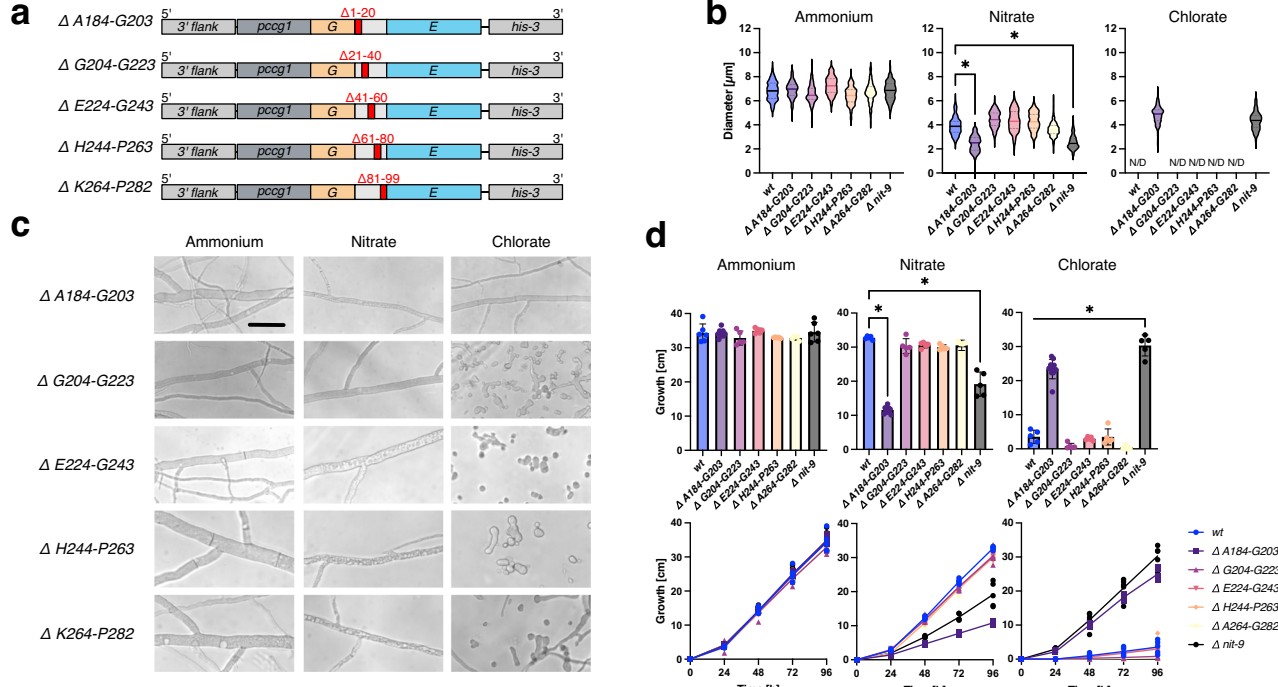

**Fig. 4 | Truncation of the Mo insertase linkage region from *N. crassa* reveals a 20 amino acid region important for nitrate-dependent growth. a** Abbreviated schematic of truncated NIT-9 variants expressed by the *ccg1* promoter in the *his-3* locus of an *N. crassa nit-9* knock-out strain. Strains are named by truncated sections. **b** Diameter of *N. crassa* main hyphae was measured after 24 h of growth on Vogel's medium (MM) agar plates (*n* = 200, 172 (Δ *E224-G243* on nitrate medium), mean and 25th–75th percentiles, *P < 0.001, one-way ANOVA). N/D non-detectable growth. **c** Hyphae of *N. crassa*, analyzed using bright-field microscopy,

bar indicates 10 µm. **d** Race tubes were grown at 30 °C in the darkness; the tubes were marked every 24 h. Growth was measured over 96 h. Experiments were performed in six independent replicates (*n* = 5–6 mean ± SD, *P < 0.01, one-way ANOVA). **b–d** All fungi were cultivated with either 80 mM ammonium or 80 mM nitrate as the sole nitrogen source; for samples with 300 mM chlorate, 25 mM ammonium nitrate was used as a nitrogen source. *wt* wild-type strain 74-OR23-1 V. Δ *nit-9 nit-9* knock-out strain.

the domains with CNX1 crystal structures determined previously by us[20,31] (Fig. 5c, d).

Modeling the convergently evolved linkage regions showed that reversing the fungal orientation also develops a short helical structure, as seen for the native NIT-9 at the N-terminus (Supplementary Fig. 8). When introducing the short 11 amino acid plant linkage region into NIT-9 in the mammal and fungi orientation (pla) a short helical structure within the linkage region is formed, keeping the domains distant from each other. In the reverse orientation (pla-r), the linkage region was modeled as a long, unstructured sequence that may be flexible enough to bring those domains into proximity. For the mammal linkage region (mam), only the reverse orientation (mam-r) showed a structured area next to the G domain. Furthermore, the mammal and fungal linkages in both orientations should not hamper interactions between the G and E domains due to the long unstructured linkage region. All runs except the native form of NIT-9 resulted in a long, unstructured linkage region. AlphaFold modeled the unstructured regions of the NIT-9 variants with a very low predicted local distance difference test (pLDDT) < 50%, while A184-G203 was modeled with a confident score (74%).

The results emphasize the complexity and evolutionary adaptation of the Mo insertase, suggesting that both the structured and unstructured regions of the linkage region are pivotal for its role in Moco biosynthesis and cellular vitality.

## Truncation of A184-G203 does not influence oligomerization prediction for NIT-9

We used structural modeling to determine the importance of the A184-G203 residues in the Mo insertase complex formation. Previous studies have shown that the activity of the E domain relies on dimerization[55], necessitating a dimerized model for accurate structural analysis (Supplementary

Fig. 9a). Additionally, when the G domain is expressed separately, it forms a trimer, with the C-terminal part of the protein not facing the trimer interface (Supplementary Fig. 9b). Based on these findings, the smallest possible oligomerization unit is a hexamer.

We modeled a NIT-9 hexamer and a Δ A184-G203 hexamer to simulate this unit. We considered a model confident if the overall pLDDT score exceeded 70% and the predicted template modeling (pTM) score was above 0.35. We expected the pTM to be low due to the large portion of the unstructured linkage region. The resulting model for native NIT-9 (pLDDT: ~74.00, pTM: ~0.37) consists of six polypeptide chains, forming a hexamer. The G domains organize into two trimeric structures on opposite sides of the complex, while the E domains from adjacent polypeptides align centrally in an arrangement of E domain dimers. This results in a hexamer with peripheral G domain trimers and a central trimeric structure composed of E domain dimers. (Fig. 6a). The unstructured part of the linkage region did not form specific structures. The A184-G203 residues formed a helical structure at the G domain trimers of the protein complex, facing outward. Residues H186, V190, and K193 face toward the dithiolene motif of MPT within the neighboring G domain (Fig. 6b, Supplementary Fig. 9c, d). *N. crassa* H186 is conserved among all species where Mo insertase is connected via a linkage region. Δ A184-G203 produced a similar model (pLDDT: ~74.00, pTM: ~0.37), comparable to the arrangement with the full-length linkage region, while all residues of the linkage region face away from MPT (Supplementary Fig. 10a, b). As for the monomers, the unstructured area of the linkage region consistently failed to achieve a pLDDT score above 50% during complex modeling.

Furthermore, we modeled the complexes with the plant and mammal linkage regions. Here, the plant model buried the linkage sequence in the middle of the protein complex, arranging two G domain trimers on the same side, contrasting the mammal construct, which was

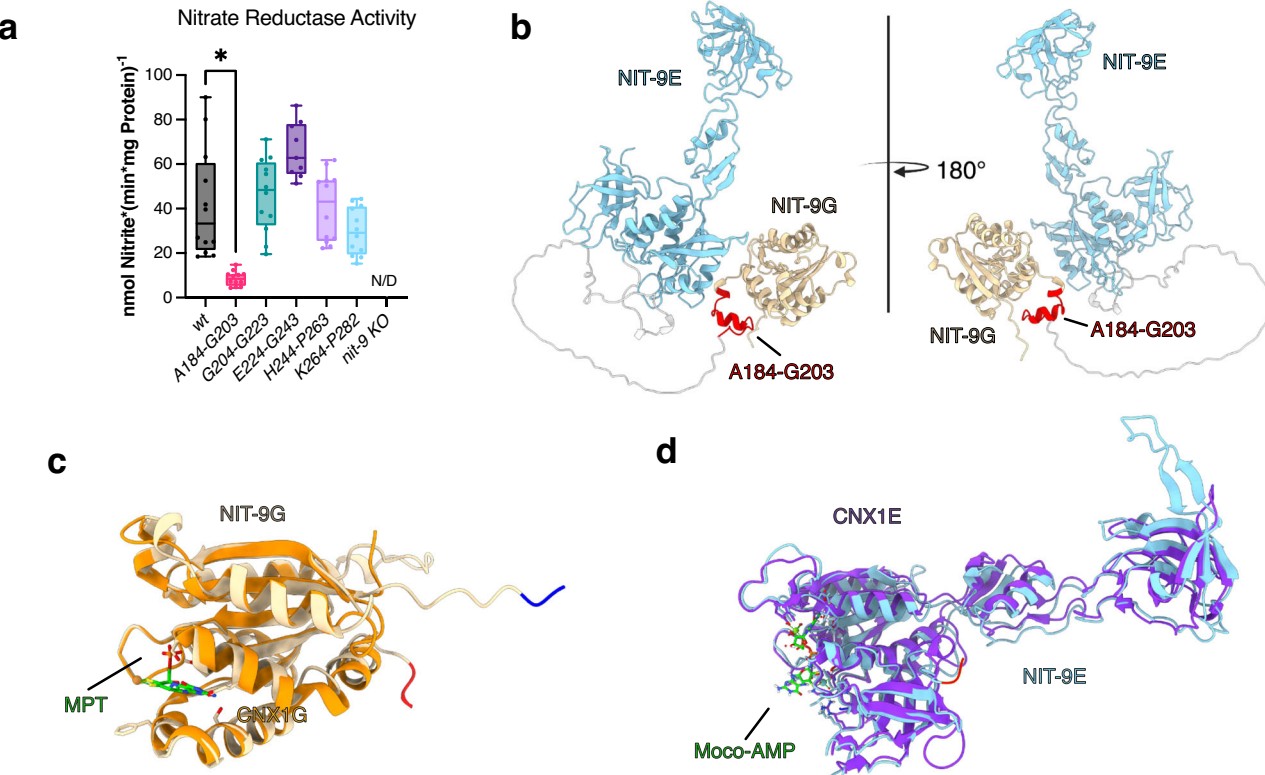

**Fig. 5 | Mo insertase depends on A184-G203 to reconstitute nitrate reductase activity in *N. crassa*. a** Nitrate reductase activity in *N. crassa* crude extracts. Strains were grown for 16 h in liquid MM with 80 mM ammonium and subsequently transferred into liquid MM with 80 mM nitrate for 3 h. Experiments were performed in independent quadruples, where each replicate comprised three technical replicates ($n = 12$, Mean, Tukey, *$P < 0.001$, one-way ANOVA). N/D non-detectable activity. **b** AlphaFold 3 generated protein structure of native NIT-9. The E domain is

depicted in sky blue, the G domain in beige, the unstructured linkage region in gray, and the structured linkage region residues in red. **c** NIT-9G domain superimposed with MPT bound CNX1G crystal structure (PDB: 1UUX) N-terminus is blue, and C-terminus is red. **d** NIT-9E domain superimposed with CNX1E crystal structure complexed with Moco-AMP (PDB: 6Q32), N-terminus is blue, and C-terminus is red.

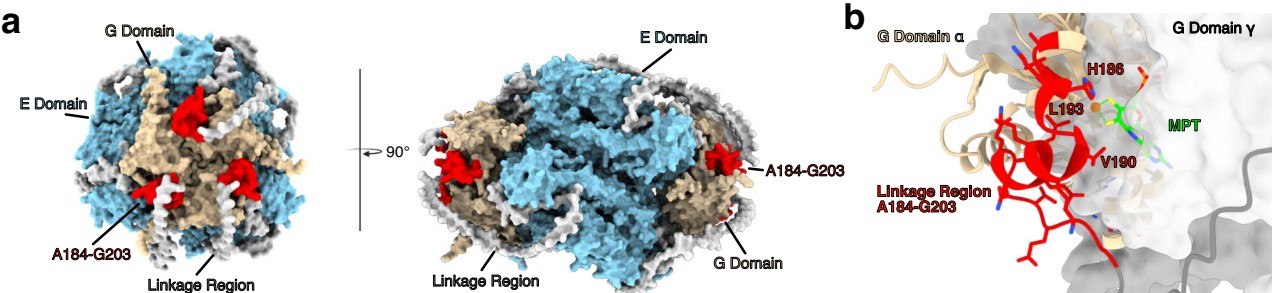

**Fig. 6 | NIT-9 hexameric complex structure reveals critical A184-G203 region. a** AlphaFold 3 models of NIT-9 hexamer. NIT-9 protein complex colored according to protein domain. NIT-9E sky blue, NIT-9G beige, the unstructured linkage region in gray, and A184-G203 in red. **b** Close view of structured linkage region of NIT-9.

Two G domains are shown. G domain γ (surface, gray) was superimposed (PDB: 1UUX) to reveal the binding position of MPT. G domain α is depicted as a cartoon structure in beige. Linkage region A184-G203 is shown in red. All residues are shown, and interface residues are highlighted.

closer to the fungal orientation, with G domains at the poles and more compressed E domain dimers in the center (Supplementary Fig. 11a, b). Additionally, we modeled CNX1 from *A. thaliana*, Gephyrin from *H. sapiens*, and MogA and MoeA from *E. coli* (Supplementary Fig. 12a–c). Unlike MogA and MoeA, the structural prediction from the Gephyrin and CNX1 model did not result in an interpretable structure with two G domain trimers.

Understanding the structural nuances is essential for unraveling the functional implications of Mo insertase in Moco biosynthesis and cellular processes.

## Discussion

Moco biosynthesis is essential for various cellular functions across eukaryotic organisms[56]. The incorporation of Mo into MPT to form Moco is a critical step catalyzed by the Mo insertase[57]. While this process is well-characterized in many organisms, the structural and functional diversity of Mo insertases across species remains intriguing. Our study aimed at exploring the evolutionary gene fusion of the Mo insertase, characterized by a highly variable linkage region, and its impact on fungal vitality.

Our comparative analysis of Mo insertases from all domains of life revealed significant diversity in sequence and domain organization. Bacteria

and archaea possess separate genes for the G and E domains[28], whereas eukaryotes display diverse arrangements, including fused or separate domains[6]. Notably, our study identified distinct linkage sequences in fungi, plants, and animals, with variations in length and composition. Gene fusion, a common process among metabolic enzymes, often enhances catalytic activity, particularly in proteins involved in complex formation or exhibiting protein–protein interactions[58,59].

To elucidate the functional significance of Mo insertase fusion, we engineered recombinant *N. crassa* strains expressing 18 variants of the Mo insertase. Substituting the linkage sequence of Mo insertase resulted in distinct phenotypic outcomes on various nitrogen sources. Strains expressing NIT-9 with plant or mammal linkage region exhibited impaired growth on nitrate medium but increased growth on chlorate medium. Conversely, the fungal linkage sequence in its original orientation under the *ccg-1* promoter complemented the *Δ nit-9* strain, surpassing *wt* growth and vitality on nitrate medium. Strains incapable of using nitrate as a nitrogen source displayed high vacuolization, facilitating cell elongation[60]. Additionally, strains with reverse orientation, a random linkage sequence, or full-length Gephyrin could not recover *Δ nit-9*. These results indicate that fungi may have independently developed their specific Mo insertase gene fusion with unique linkage sequences.

The unstructured region in the linkage sequence between fused metabolic enzymes often arises due to genetic rearrangements or mutations during evolution[38,61]. The 49% sequence identity of the linkage region between the filamentous fungi *A. nidulans* and *N. crassa* suggests that this gene fusion event is not unique to *N. crassa* and may have occurred earlier in fungal evolution. Further experiments revealed that the separate expression of Mo insertase domains, as seen in lower eukaryotes, bacteria, and archaea, could not rescue the *Δ nit-9* strain, even though all catalyzing enzymes for Moco biosynthesis were present[62]. This led us to explore protein fragment complementation using heterokaryotic strains encoding separate genes for the G or E domains fused with the fungal linkage sequence and the E or G domain, respectively. Only *G-linker + E* showed minimal recovery of Mo insertase activity, indicating that protein complementation is only achievable if the linkage region is truncated distant from the G domain. When, however, the structured part of the linkage region is disrupted, no complementation is achieved. The low recovery observed aligns with previous in vitro experiments for Gephyrin, which showed a 300-fold increase in activity for the fused domains compared to separate expression of the G and E domains[26].

Step-by-step truncation analysis of the linkage region revealed a 20-amino acid sequence (A184-G203) crucial for fungal growth on nitrate medium. Strains with this truncation exhibited phenotypic similarities to the *Δ nit-9* strain, indicating a loss of Mo insertase activity. In vitro NR assays confirmed the importance of this region in Moco biosynthesis. Structural modeling of the NIT-9 Mo insertase with fungal, mammal, and plant linkage regions provided insights into their organization and potential functional roles. The boundaries of the NIT-9 G and E domains are distant from the active site, so truncating them should not influence the activity or oligomerization of these domains. The predicted hexamers showed that the unstructured linkage region could not be reliably modeled; however, the A184-G203 sequence was predicted confidently.

The inability to detect MPT-AMP or MPT/Moco in the truncation strains, despite NR activity, suggests that the Moco biosynthesis intermediates are well below a detectable threshold in *N. crassa*. While no quantifiable amounts were observed, the results for A184-G203 indicate a reduction in Moco production, which may account for the diminished NR activity. Moco is extremely oxidation-sensitive[6], suggesting that Mo insertase must have tight interaction and significant protein movement. The G domain must receive MPT from the MPT synthase complex, and the E domain must transfer newly formed Moco to the Moco-dependent enzymes. Structural models indicate that the active site of the G domain faces another G domain and the structured part of the linkage region. Here, H186, V190, and K193 are in proximity to the dithiolene motif. The truncation of the A184-G203 region or splitting NIT-9 into two proteins

completely abolished fungal growth, highlighting the critical nature of these residues. Nevertheless, previous in vitro experiments have shown that the G domain of Gephyrin and CNX1 are catalytically active without the N-terminal part of the linkage region or the unspecified sequence following CNX1G[18,26].

Thus, we conclude that A184-G203 is crucial not for catalytic activity or complex formation but for other intracellular processes, such as product-substrate channeling or protein-protein interaction, ensuring the safe transfer of MPT or MPT-AMP. This hypothesis has also been previously made for the linkage region of Gephyrin in mammals[26]. Additionally, previous studies have shown that CNX1 and Gephyrin bind to the cytoskeleton, indicating that the linkage sequence may be independently important for anchoring the Mo insertase[23,63]. In humans, Gephyrin is essential for clustering GlyR and GABA receptors at the inhibitory synapse, a process mediated by the underlying tubulin and actin cytoskeleton[64]. NIT-9 potentially forms a hexameric complex or a mesh-like structure, as proposed in mammals[65,66], further emphasizing the functional significance of the linkage region.

These findings open avenues for further exploration using more sensitive assays or alternative approaches to pinpoint the specific step in the biosynthetic pathway affected by the Δ A184-G203 mutation. Although the MPT-AMP and MPT/Moco levels were undetectable, the significant reduction in NR activity suggests that Moco biosynthesis is disrupted at a critical stage, without being completely abolished. Further research will be essential to fully unravel how this section is crucial for Moco production and its broader impact on NR function. Moreover, understanding the unique traits of the convergently evolved Mo insertase fusion will offer valuable insights into enzyme evolution, potentially revealing novel mechanisms of functional adaptation in complex biosynthetic pathways.

## Methods

### *N. crassa* strains
As wild-type strain *N. crassa* 74-OR23-1 V; *mat A* (FGSC2489) was used. For generation of *Δ nit-9, his-3*; *mus-52::bar*; *mat A* (FGSC988) and *NCU09746::hph; mus52::bar;* *mat a* (FGSC18574) were crossed. See Supplementary Table 1 for all *N. crassa* strains. Non generated strains were obtained from the Fungal Genetic Stock Center (https://www.fgsc.net).

### *N. crassa* handling
*N. crassa* liquid cultures were cultivated at 130 rpm. Strains were grown at 30 °C on Vogel's minimal medium (MM)[67], supplemented with 1.5% (w/v) sucrose as a carbon source. If required, 1.5% (w/v) low melting agarose (Carl Roth Karlsruhe, Germany) was used to solidify the medium. Appropriate supplements and antibiotics were added for strains carrying resistance markers for selection. For growth under Moco-dependent or Moco-independent conditions[14], the MM was modified, and ammonium nitrate was removed from the recipe. As a replacement, 75 mM ammonium chloride ($NH_4Cl$) or 80 mM potassium nitrate ($KNO_3$) were added to MM. For strains grown on chlorate medium, 300 mM sodium chlorate ($NaClO_3$) was added to MM.

### Crossing of *N. crassa*
The FGSC18574 strain was incubated on a synthetic cross-medium[68] at room temperature until protoperithecia started to form. The conidia of FGSC988 were harvested after 3–5 days of growth in slant agar tubes at 30 °C. The mating partner was evenly distributed over the protoperithecia. The crossing plates were incubated at room temperature until ascospores were ejected. The ascospores were incubated for ~7 d in ripening buffer (0.1 M Tris-HCl pH 8.0, 2 mM EDTA) in the dark at room temperature. An ascospore suspension was heat-activated for 20 min at 60 °C and grown on BDES plates[69] at 30 °C overnight to germinate the spores.

### *N. crassa* genotyping
Mating type determination was carried out as described in the crossing section by crossing the newly germinated strains with a mating type A and

Mating type a strain[70]. For the genotyping, MM was used for growth under standard ammonium nitrate conditions. For growth on nitrate MM, 80 mM potassium nitrate was added as the sole nitrogen source. 300 mM sodium chlorate was added to MM for the chlorate plates. $10^5$ conidia were used to inoculate slants, which were grown for two days in the dark at 30 °C and one day at room temperature in a day and night light cycle.

## Transformation of *N. crassa* strains

*N. crassa* strains were transformed by electroporating macroconidia using a Gene Pulser Xcell Electroporation System (Bio-Rad, Hercules, USA)[71]. For preparation, the strains were cultivated in 500 ml flasks containing 50 ml of solid MM at room temperature for at least 7 days. Conidia were harvested with 40 ml of 1 M sorbitol, filtered through cheesecloth, and centrifuged at $3500 \times g$ for 3 min. The supernatant was discarded, and the conidia were washed twice more with 45 ml of 1 M sorbitol. After decanting the final wash, the conidia were resuspended in the remaining sorbitol. For electroporation, 100 µl of the conidia suspension was transferred to a 1 mm electroporation cuvette and pulsed at 1500 V, 600 Ω, and 25 µF for 5 ms. Immediately following electroporation, 1 ml of 1 M sorbitol was added to the cuvette, and the conidia were mixed with 30 ml of pre-warmed top agar (MM, 1% agar, 1× FIGS). The mixture was plated in 5 ml portions onto bottom agar (MM, 1.5% agar, 1× FIGS) plates. Transformed conidia were incubated in the dark at 30 °C for 3–4 days. Germinated conidia were then harvested and transferred into slants containing 5 ml of MM. Homokaryotic strains were obtained by single spore isolation[41]. Afterward, DNA was isolated, and the strains were verified via PCR. Integration of the constructs into the *his-3* locus was confirmed by two separate PCR reactions: (1) Homologous recombination, primed at *his-3* and the gene of interest; (2) Homokaryon screening, primed at *his-3* and the *wt* flank removed during recombination. All integration PCR fragments were sequenced via Sanger sequencing (Microsynth Seqlab Göttingen, Germany). The PCR for verification of homokaryons showed no bands for any of the strains tested, confirming their homokaryotic nature. Primers are listed in Supplementary Table 2.

## Race tube assay with *N. crassa*

For the race tubes, approx. 45 cm long glass tubes with ends bent upwards at an angle were used. They were sealed with lids and sterilized using an autoclave. 8 ml of sterile MM with ammonium, nitrate, or MM with 300 mM chlorate were filled in the tubes. Conidia were harvested from a slant agar tube, and their cell density was determined using a photometer. The suspension was diluted to $10^6$ conidia per ml ($OD_{530} = 0.1 \triangleq 2 \times 10^5$ conidia/ml). The race tubes were inoculated with $10^4$ conidia in 10 µl. The starting point was marked, and the race tubes were incubated for 96 h at 30 °C in the dark. Growth was measured every 24 h.

## Cultivation of *N. crassa* for Western blot analysis and NR assay

Conidia were harvested with 25 ml ultrapure $H_2O$ (MilliQ Merck Darmstadt, Germany) and filtered via cheesecloth. Additionally, the conidia were washed with 50 ml ultrapure $H_2O$, centrifuged for 5 min at $4500 \times g$, and resuspended in 5 ml. The OD at 530 nm was measured to determine the number of conidia. An Erlenmeyer flask containing 500 ml of liquid MM with 80 mM ammonium as a nitrogen source was inoculated with $10^5$ conidia per ml for 12–16 h at 28 °C, shaking at 130 rpm. The mycelium was harvested using Mira cloth (Merck Millipore Burlington, USA). The pellet was washed 3 times with 10 ml ultrapure $H_2O$ and transferred into Erlenmeyer flasks containing 50 ml liquid MM containing 80 mM nitrate as a nitrogen source. The mycelium was harvested after 16 h of incubation at 30 °C, shaking at 130 rpm. After harvest, the mycelium was flash-frozen in liquid nitrogen.

## Microscopy of *N. crassa* hyphae

To assess the hyphal morphology of Mo insertase variants, macroconidia were harvested from slants using 1 ml of ultrapure $H_2O$ and filtered through cheesecloth. A total of $10^4$ conidia were used to inoculate petri dishes, which

were incubated at 30 °C for 12 h. A piece of agar from the growth front was excised and examined using a Nikon Eclipse Ni upright microscope equipped with a DS-Qi2 camera (Nikon, Tokyo, Japan)[72]. Imaging was performed with a 60x Plan Apo VC objective lens (1.4 numerical aperture, oil immersion). The images were captured and processed using NIS Elements software (Nikon, Tokyo, Japan) to document the hyphal structures[14].

## Isolation of Mo insertase genes

After overnight growth of *N. crassa* mycelium in a liquid MM, the total RNA or DNA was extracted. The mycelium was frozen in liquid nitrogen, and 500 µl of ice-cold DNA/RNA-Shield (Zymo Research Freiburg, Germany) was added to 200 mg mycelium. Subsequently, the mycelium was lysed in a FastPrep-24 (M.P. Biomedicals Irvine, USA) four times for 30 s at 6.5 m/s with 5-min breaks and then incubated for 5 min on ice. According to the manufacturer, RNA was further isolated using the Quick-RNA Fungal/Bacterial Kit (Zymo Research Freiburg, Germany). DNA was further isolated using Phenol/Chloroform extraction[14]. For cDNA synthesis, the ProtoScript® II First Strand cDNA Synthesis Kit (New England Biolabs Ipswich, USA) was used according to the manufacturer's protocol. The *nit-9* CDS (NCU09746) was amplified via PCR for the linkage sequences from fungi. For the linkage region of mammals, MOCOS, as well as full-length gephyrin, a human cDNA Library from the liver (Merck Millipore Burlington, USA) was used. The linkage region from plants was introduced via site-directed mutagenesis.

## Seamless cloning and mutagenesis

The seamless cloning method was performed with the NEBuilder® HiFi DNA Assembly Cloning Kit (New England Biolabs Ipswich, USA). As a shuttle vector for multiplying plasmids in *E. coli* (NEB5α New England Biolabs Ipswich, USA) and genomic integration of DNA fragments into the *N. crassa his-3* locus, pMF272 was used[37]. The DNA assembly was carried out as described in the manufacturer's manual, but the reaction volume was constantly reduced to a minimum. The plasmids were directly transformed into *E. coli* after the DNA assembly. Nucleotide exchanges were performed with the Q5® Site-Directed Mutagenesis Kit (New England Biolabs Ipswich, USA). Q5® High-Fidelity DNA Polymerase was used to verify the inserted genes. (New England Biolabs Ipswich, USA). All primers and methods for the created constructs are listed in Supplementary Table 2. Oligonucleotides were ordered from Merck (Darmstadt, Germany).

## Protein extraction from *N. crassa* crude extracts

2 ml screw-cap tubes were filled with 0.5 ml of 0.5 mm and 0.1 mm (mixing ratio 3:2) zirconium oxide beads, 800 µl *N. crassa* protein extraction buffer (50 mM PBS, 200 mM NaCl, 1 mM EDTA, 1 mM PMSF, 1 µg/ml Leupeptin, 1 µg/ml Pepstatin A) and stored on ice. Subsequently, flash-frozen mycelium was hit with a hammer to crush it. 200–300 mg of mycelium was transferred into the pre-cooled 2 ml tubes. The cell disruption was performed with three runs in the FastPrep-24 (MP Biomedicals Santa Ana, USA) at 6.5 m/s for 30 s with 5 min of cooling on ice in between. After lysis, the samples were stored on ice for 20 min. Afterward, they were centrifuged at $21,100 \times g$ for 40 min at 4 °C. The supernatant was transferred to a new reaction tube and stored on ice until use for further analysis.

## Western blot analysis

For the detection of the expressed proteins introduced into *N. crassa*, Western blot analysis was performed. We used a monoclonal antibody from murine hybridoma cells targeting the NIT-9E domain generated by the Antibody Facility (BRICS Braunschweig, Germany). The immunoblot was carried out following a 12.5% SDS-PAGE in which 50 µg crude extract was loaded into each lane. The proteins were transferred to a polyvinylidene fluoride (PVDF) membrane using the semi-dry Trans-Blot Turbo Transfer System (Bio-Rad Hercules, USA). The membranes were then blocked for 1 h in 5% non-fat dry milk (NFDM) in TBS-T to prevent nonspecific binding. Primary antibody (1:1000 of a 1 mg/ml solution) was added for 1 h incubation with agitation. The membrane was washed three times for 5 min

with TBS-T. Subsequently, a 1 h incubation with an HRP-conjugated secondary goat anti-mouse antibody (1:20,000; Cat# 31430, RRID AB 228307, Thermo Fisher Scientific Waltham, USA) in 5% NFDM was performed in TBS-T, followed by three additional washes. According to the manufacturer's instructions, the PVDF membrane was visualized with SERVA-Light Eos CL HRP WB Substrate Kit (SERVA Electrophoresis Heidelberg, Germany). The specific protein bands were detected and documented using the ChemiDoc XRS+ imaging system (Bio-Rad Hercules, USA).

### Nitrate reductase assay

The nitrate reductase (NR) assay was conducted using a calibration curve ranging from 0.00025 mM to 100 mM for nitrite quantification[52]. *N. crassa* crude extracts were diluted to a protein concentration of 0.67 µg/ml. Each sample, consisting of 30 µl of standard or 20 µg of crude extract, was combined with 30 µl of NR mix 1 (3 µl 20 mM NADPH, 6 µl 100 mM KNO$_3$, 3 µl 1 mM FAD, and 18 µl extraction buffer) in 1.5 ml reaction tubes. After incubating for 10 min in the dark, the reaction was halted by adding 40 µl of 0.6 M zinc acetate. Following this, 400 µl of NR mix 2 (1% sulfanilamide in 3 M HCl, 0.02% 1-naphthylamine) was added, and the samples were incubated for an additional 10 min in the dark at room temperature. The samples were thoroughly mixed at each step and vortexed. After centrifuging at 12,000 × *g* for 12 min at room temperature, 200 µl of the supernatant was transferred to a 96-well plate, and the absorbance was measured at 540 nm using a Multiskan GO plate reader (Thermo Fisher Scientific, Waltham, USA).

### MPT-AMP and Moco/MPT quantification in *N. crassa* crude extracts

Crude protein extracts (800 µg in 400 µl) were oxidized by adding 50 µl of an acetic iodine solution (1% (w/v) iodine, 2% (w/v) potassium iodide in 1 M HCl) and incubating for 16 h at room temperature in the dark. After oxidation, 56 µl of 1% (w/v) ascorbic acid was added to neutralize the reaction, followed by 200 µl of 1 M Tris (unbuffered) and 13 µl of 1 M MgCl$_2$. For MPT-AMP quantification, 1 µl of phosphodiesterase I (1 U Merck Darmstadt, Germany) was added, while for MPT/Moco quantification, 1 µl of calf intestine alkaline phosphatase (1 U, Roche Basel, Switzerland) was added, and both reactions were incubated overnight at room temperature. After this, MPT-AMP samples underwent an additional overnight incubation with 1 U of alkaline phosphatase. Following incubation, the samples were centrifuged at 16,000 × *g* for 20 min and filtered through a 0.2 µm filter prior to HPLC injection. Detection of dpFormA was carried out at room temperature using a reversed-phase C-18 column (250 mm × 4.6 mm, 5 µm, ReproSil-Pur Basic C-18 HD) on an Agilent 1100 HPLC system equipped with a fluorescence detector. The chromatography was performed at a flow rate of 1 ml min$^{-1}$ using an isocratic run with 5 mM ammonium acetate and 15% (v/v) methanol as the mobile phase. Fluorescence detection was set at $\lambda_{ex} = 302$ nm and $\lambda_{em} = 451$ nm. Data analysis was conducted using OpenLab CDS Version 2.2.0.600, with synthetic dpFormA used for calibration[27,57]. For each strain 5–9 biological replicates were tested.

### Evolutionary analysis by phyloT, iTOL, MEGA11, and BLAST search

Evolutionary analyses were conducted in phyloT and iTOL[73]. The evolutionary history of Mo insertases was accomplished using the NCBI data bank. Mo insertase genes from *N. crassa* were used in BLAST to identify other organisms containing Mo insertases[74]. MEGA alignments and tree generation were performed using the sequences from the BLAST search. The Maximum Likelihood and JTT matrix-based models were used for the evolutionary history. Initial trees for the heuristic search were obtained automatically by applying Neighbor-Join and BioNJ algorithms to a matrix of pairwise distances estimated using the JTT model, and then selecting the topology with superior log likelihood value. Evolutionary analyses were conducted in MEGA11[75]. All insertases accession numbers are listed in Supplementary Table 3.

### Multiple sequence alignment using SnapGene

In addition to the analysis in phyloT and iTOL, all sequences for the full-length Mo insertases, as well as the E and G domains acquired from NCBI, were aligned in SnapGene software using MAFFT (from Insightful Science; available at snapgene.com) to highlight conserved and important amino acids.

### Structural prediction using AlphaFold 3

AlphaFold 3 was utilized using the DeepMind AlphaFold servers provided by Google to analyze the various motifs of the targeted Mo insertase protein[54]. The amino acid sequences of the construct CDS predicted in SnapGene were used as sequence queries. The model was run for six cycles, and the highest-ranked models were used to visualize the structure in ChimeraX[76].

### Statistics and reproducibility

Results from the microscopic assays were collected by measuring hyphae within 20–30 microscopic fields generated from at least three independent replicates. At least 6 tubes were used for each strain in the race tube assay. Each tube was inoculated from independently isolated homokaryotic conidia. Isolated conidia that did not germinate (nitrate and ammonium medium) were excluded from the measurements. In the NR assay, four independently cultured mycelia were harvested and lysed for each strain. Protein was extracted, and NR assay was conducted three times for each mycelium. Data are plotted as means and 25th–75th percentiles in the violin plots used for hyphal diameter. Statistical significance (*P* values) was calculated using ordinary one-way ANOVA with a family-wise alpha threshold and confidence level of 0.001 in GraphPad Prism 10.1.1. Data are plotted as means ± standard deviation (SD) for the bar graphs used for the race tube assay. Statistical significance (*P* values) was calculated using ordinary one-way ANOVA with a family-wise alpha threshold and confidence level of 0.01 in GraphPad Prism 10.1.1. Data are plotted as means in the box plot for the NR assay, and Tukey whiskers are used. Statistical significance (*P* values) was calculated using ordinary one-way ANOVA with a family-wise alpha threshold and confidence level of 0.001 in GraphPad Prism 10.1.1.

### Reporting summary

Further information on research design is available in the Nature Portfolio Reporting Summary linked to this article.

## Data availability

All data supporting the findings of this study are included within the manuscript and its Supplementary Information. The numerical source data used in this research can be found in Supplementary Data 1 and are available upon reasonable request from the corresponding author. Additionally, uncropped images of gels and blots are provided in Supplementary Figs. 13 and 14. The plasmids created for transformation of *N. crassa* strains are deposited at Addgene (https://www.addgene.org) under ID228409–228426. All other data can also be obtained from the corresponding author upon reasonable request.

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

## Acknowledgements

Funding for this research was provided by a grant from the Deutsche Forschungsgemeinschaft [GRK 2223/1]. We express our profound gratitude to all members of the Mendel lab and colleagues from the TU Braunschweig Department of Plant Biology for their valuable comments and engaging discussions.

## Author contributions

K.D.O. designed the research. M.R., E.H., K.S., and K.D.O. performed the research. M.R., E.H., and K.D.O. analyzed the data. Conceptualization: K.D.O. Methodology: M.R., E.H., K.S., and K.D.O. Formal analysis: M.R., E.H., K.S., and K.D.O. Writing—original draft: K.D.O. Writing—review & editing: R.H., R.R.M., and K.D.O. Resources: R.H., and R.R.M. Visualization: K.D.O. Project administration: R.R.M., and K.D.O.

## Funding

## Competing interests

The authors declare no competing interest.
