## [Transparent Peer Review file · Communications Biology]

Convergent evolution links molybdenum insertase domains with organism-specific sequences

Corresponding Author: Dr Kevin Oliphant

Version 0:

Reviewer comments:

Reviewer #1

(Remarks to the Author)

The manuscript by Rabenow et al. investigates the Molybdenum insertase from *Neurospora crassa*. They demonstrate that substituting or reorienting the linkage region with convergently evolved sequences from other species, such as mammals and plants, revealed a Moco deficient phenotype. Furthermore, stepwise truncation analysis and structural modeling unveiled a crucial 20-amino acid sequence within the linkage region essential for fungal growth. In their work, Rabenow et al. mainly used a *N. crassa* mutant strain and complemented that strain with different constructs expressing the G, E or linker region or a combination from different organisms. The positive effect of the complementation was mainly tested by analyzing the growth of the strain on medium with nitrate or chlorate. This is not a good method to quantify the efficiency of the complementation (even measuring the wet weight would be more quantitative), which rather should be quantified by measuring the activity of the nitrate reductase enzyme (in Units per mg or kcat). Further, the authors investigate the importance of the linker region by analyzing truncated versions of it. This has already been reported for gephyrin by Belaidi and Schwarz in 2013, so the novelty is limiting. Further, more descriptions are required in what is already known about gephyrin and the linker region, both in the introduction and in the discussion for comparison.

Also, what would be really interesting is to dissect which step in Moco insertion is hampered in the mutants, MPT_AMP formation or molybdenum insertion. Both can be separated and quantified (by formA-AMP, formA or nit-1 reconstitution) and would give novelty to the work, The dissection of the step on which the linker region has an influence would be really interesting and justify publication in a journal like nature communications. Otherwise, I recommend publication in a lower ranking journal that does not require quantification of enzyme activities or Moco intermediates and is happy with just growth on nitrate.

Reviewer #2

(Remarks to the Author)

I found this to be a very interesting paper. The experiments were rigorously done and well explained. In some ways, it was a very esoteric topic, looking at the linker region that join two genes. My own work is not focused on molybdenum biochemistry, but I found the results of this paper to be a general interest. These data provide an important addition to our understanding of protein evolution. *Neurospora* was a good model organism for addressing these questions because of the way the function of nitrate reductase could assayed - required for growth in some conditions, but not others. I also appreciated the modeling studies included in this paper. As I read the first part of the paper, I was thinking that they might be able to use the new alpha fold technology for further analysis of their data. In fact, that's what they did. I have no suggestions for substantive changes in the paper.

My suggestions for minor changes are as follows:

Line 106. Don't use "homology" as a shorthand for "sequence similarity." Genes and proteins can be homologous but amino acids are not homologous.

Line 114 "sequence that is insignificant for binding MPT". Do you mean that this region is not involved in binding MPT?

Line 122 This heading should be rewritten. You want to say something like *N. crassa* cannot grow on nitrate if the Mo invertase linkage region is altered.

Line 259 -261. I had trouble understanding what this is trying to describe.

Line 278 to end of results. I think this section on modeling is very interesting and is an important part of the paper. However, I

read it several times and struggled to understand the structures. For example, 288-289, the whole complex is a hexamer, but has “three E domain dimers with G domain trimers.” Is there a trimer of dimers? How many polypeptides? I realize it is complicated, but other readers may also struggle with this. It needs a bit of rewriting.

Reviewer #3

(Remarks to the Author)

This work follows investigations of this group on Moco biosynthesis. The present study focuses on the linker region that links the G to E domains of Mo insertase in some organisms (like some animals and fungi), whereas it sometimes connects E to G domains (like in plants) or is even absent in some organisms that use separate proteins (bacteria, archaea). The authors investigated the role of this linker region on the fungal enzyme, by substituting it with several variants:

- using the sequence found in plant (*A. thaliana*), mammal (*H. sapiens*) or native (*N. crassa*), expressing either the domains in the native order (as of fungal origin; ie G-linker-E) or reversed (E-linker-G)
- removing the linker or separating E and G domains, with the linker either suppressed or added after the E domain (E-linker) or before the G domain (linker-G)
- using the native sequence that is deleted from stretch of 20 aminoacids covering the linker region.

Using different assays (hyphae diameters, growth in race tubes, eventually nitrate reductase activity) the authors are able to obtain a phenotype of all the constructs in different culture condition (ammonium, nitrate and chlorate) and compare it with WT and a NIT-9 mutant. These different constructs were also modelled using AlphaFold3, giving some clues about the role of the linker.

This work allows the authors to propose that this linker region convergently evolved in eukaryote, with the importance of specific parts of the linker in protein function.

The experimental part of this work is well conducted, with experiments leading to clear conclusions. On the other hand, the modelling part is sometime confusing, with too much description, especially when the level of precision of the models does not allow for this level of detail. The parts of the paper describing structural model should therefore be reviewed, reducing the number of detailed analyses and retaining only the broad outlines.

In the end, I'm sorry to find that the conclusion is rather weak.

Here are more specific comments:

- In combination with part b of Figure 1 I would be interested in a phylogenetic tree of the individual E and G domains. That could be assembled in an additional figure and, depending on the trees obtained, it could add a small context on the evolutionary history of these domains.
- Sometime the authors refer to percentage of sequence homology (l. 106, 107, 306), although this notion is not defined (which amino acid is homologous to which other Aa). Sequence identity should be used instead, as there is no ambiguity.
- Supp Fig. 3a is difficult to follow without context, I would recommend either to refer to Fig. 2a and Fig. 3a, or to repeat these panels in a new Supp Fig. 3b panel.
- I think there are too many figures representing structural models in the main figures (Fig. 5 b-e; Fig. 6a-e). I recognise that AlphaFold (AF) is an extraordinary tool for analysing protein sequences, but the level of detail in the analyses must bear in mind that these are only models. I therefore recommend reducing the number of panels in Figures 5 (placing panels c-e in a supplementary figure, for example) and 6 (only keeping panels d and f for example).
- Discussion about the relative position of the E and G domains (l.262-273) in different species and variants is difficult to follow, especially because 1/ the oligomeric assembly is unknown (or eventually described later with Figure 6), and 2/ the scores given by AF seem rather low (although not indicated; I ran such a prediction and found a pTM of 0.63. Also, I noted that in the five models proposed by AF3 the E and G domains adopt different relative positions). I therefore recommend to explicitly indicate the pTM for each model in panel e. But then I'm not sure of what we can say about these models, except that E and G domains adopt different positions and part of the linker (in the fungal and mammal proteins) is structured.
- I agree that a hexameric structure is the minimal unit. However, a pTM of 0.37 (ipTM is approximately the same) is low (for instance, AF3 note that “ipTM measures the accuracy of the predicted relative positions of the subunits within the complex. Values higher than 0.8 represent confident high-quality predictions, while values below 0.6 suggest likely a failed prediction.”)
- Supplementary Figure 1 (important as it describes the domain and linker boundaries) seems partly wrong to me. A single alignment of the full-length proteins should be more insightful (although excluding the plant's sequence because of the inversion of domains), with a clear indication of domains and linker boundaries (if present). A separate alignment of the linkers only (including plant) might be insightful. Writing this, I realize that the equivalent of *N. crassa* H186 (obviously the candidate for an important function in the linker) has an equivalent in the C-Terminus of the plant sequence (in the “... PKHIP...”, found on panel b).
- l.289-291 are very odd (“stretched without tension”, “poles” and “outward” would need to be defined). The whole paragraph should be simplified.
- “nicked distant” (l.345) doesn't seem to be adapted.

Author Rebuttal letter:

Reviewer 1:

Referee comment Response

1 The manuscript by Rabenow et al. investigates Thank you for your comments. I would like to clarify that the Molybdenum insertase from *Neurospora* the use of race tubes to measure fungal growth is a well-

crassa. They demonstrate that substituting or established method in the field, including studies related reorienting the linkage region with convergently to Moco biology. For instance, Wajmann et al. (2020) evolved sequences from other species, such as successfully utilized this approach and was able to mammals and plants, revealed a Moco deficient measure differences in growth by single amino acid phenotype. Furthermore, stepwise truncation substitutions. Our primary objective was to assess the in analysis and structural modeling unveiled a vivo effects of the mutations, aiming to determine crucial 20-amino acid sequence within the whether these specific strains could restore wild-type linkage region essential for fungal growth. In their growth. The results were highly reproducible, with low work, Rabenow et al. mainly used a N. crassa standard deviations, and the significant differences in mutant strain and complemented that strain with growth were clearly observable in both the figures and different constructs expressing the G, E or linker the graphs. Therefore, I find it challenging to understand region or a combination from different organisms. why these results would be considered unconvincing. The positive effect of the complementation was Additionally, as shown in Figure 5a, we did measure NR mainly tested by analyzing the growth of the activity for the truncation strains. We also performed NR strain on medium with nitrate or chlorate. This is activity assays for the other strains; however, these not a good method to quantify the efficiency of assays showed no detectable activity, which is why we the complementation (even measuring the wet chose not to include them in the manuscript. weight would be more quantitative), which rather Please elaborate further on why race tubes are a bad should be quantified by measuring the activity of method to measure the response of this mutant strains the nitrate reductase enzyme (in Units per mg or as I have not seen a downside of the method so far. kcat).

2 Further, the authors investigate the importance Thank you for raising this point. I would like to clarify that of the linker region by analyzing truncated while Belaidi and Schwarz (2013) did investigate the role versions of it. This has already been reported for of the linker region in gephyrin, their focus was on gephyrin by Belaidi and Schwarz in 2013, so the different splice variants and their influence on in vitro novelty is limiting. reconstitution. They demonstrated that specific splice variants of gephyrin play distinct roles in in vitro Moco biosynthesis. However, they did not conduct truncation analysis of the linkage region.

In our study, we took inspiration from the work of Belaidi and Schwarz but aimed to explore whether their findings were applicable to fungi. Our alignment shows that the linkage regions across the observed organisms are highly diverse. Importantly, our results reveal that the linkage region from gephyrin, as well as the full-length protein, cannot reconstitute activity in fungi and were previously only described to restore growth in mammalian cells and plants. Belaidi and Schwarz did not test fungi in their study.

Furthermore, our work goes beyond their findings by pinpointing a critical 20-amino acid region within the linkage that is essential for fungal growth. We have discovered a conserved histidine within this region facing towards the dithiolene motif of MPT within the active site of the G domain only present in evolutionarily fused Mo insertases. This level of detail and the focus on a different organism highlight the novelty and significance of our findings.

3 Further, more descriptions are required in what is Thank you for your feedback. While gephyrin is not the already known about gephyrin and the linker main focus of our manuscript, we recognize the region, both in the introduction and in the importance of providing sufficient context for readers. To discussion for comparison. address this, we have added sentences in the introduction that outline the previous knowledge regarding the role of the linker region in the in vitro function of Moco biosynthesis using gephyrin, particularly referencing the work by Belaidi and Schwarz (2013).

Additionally, we have expanded the introduction to include a more comprehensive overview of gephyrin, specifically detailing its dual role in Moco biosynthesis and neuronal anchoring. In the discussion, we now offer a more thorough comparison between our findings and the established knowledge on gephyrin, highlighting the structural and functional differences across species and emphasizing the unique insights from our study, such as the identification of a critical 20-amino acid region essential for fungal growth.

These additions aim to strengthen the manuscript by providing the necessary background and context while

maintaining the primary focus on our novel findings.

4 Also, what would be really interesting is to Thank you for your suggestion. We agree that dissecting dissect which step in Moco insertion is hampered which step in Moco insertion is affected in the mutants in the mutants, MPT_AMP formation or would add valuable insights. However, performing MPT-molybdenum insertion. Both can be separated AMP and Moco/MPT analysis via FormA oxidation on and quantified (by formA-AMP, formA or nit-1 these strains is unfortunately not feasible due to the reconstitution) and would give novelty to the insufficient concentration of metabolites in N.

work, The dissection of the step on which the crassa crude extracts. We attempted to analyze FormA linker region has an influence would be really samples, but were unable to detect more than interesting and justify publication in a journal like background signal, even after purifying the extracts nature communications. Otherwise, I recommend using anion exchange columns and concentrating the publication in a lower ranking journal that does samples. These challenges prevented us from obtaining not require quantification of enzyme activities or reproducible MPT-AMP to Moco/MPT ratios.

Moco intermediates and is happy with just It is important to note that our study focuses on in vivo growth on nitrate. effects rather than in vitro analysis with purified proteins, which would have allowed for such measurements.

However, this was not the primary goal of our work. this manuscript was submitted to Communications Biology, and we believe the study's focus on in vivo growth analysis and the identification of a critical linker region is well-aligned with the journal's scope.

Reviewer 2:

I found this to be a very interesting paper. The experiments were rigorously done and well explained. In some ways, it was a very esoteric topic, looking at the linker region that join two genes. My own work is not focused on molybdenum biochemistry, but I found the results of this paper to be a general interest. These data provide an important addition to our understanding of protein evolution. Neurospora was a good model organism for addressing these questions because of the way the function of nitrate reductase could assayed - required for growth in some conditions, but not others. I also appreciated the modeling studies included in this paper. As I read the first part of the paper, I was thinking that they might be able to use the new alpha fold technology for further analysis of their data. In fact, that's what they did. I have no suggestions for substantive changes in the paper.

Referee comment Response

1 My suggestions for minor changes are as Thank you for the notion, we have addressed it as you follows: suggested

Line 106. Don't use "homology" as a shorthand for "sequence similarity." Genes and proteins can be homologous but amino acids are not homologous.

2 Line 114 "sequence that is insignificant for Yes, the CNX1G protein crystals were created without binding MPT". Do you mean that this region is this region, showing that it is not involved in binding not involved in binding MPT? MPT. I addressed your concern and added it as not necessary for binding MPT to make it clearer

3 Line 122 This heading should be rewritten. You Thank you, the heading has been updated as you want to say something like N. crassa cannot suggested grow on nitrate if the Mo invertase linkage region is altered.

4 Line 259 -261. I had trouble understanding what Thank you for your feedback. In response to your and this is trying to describe. another reviewer's comments, I have removed the entire quote from lines 259-261. This change aligns with the suggestion to reduce the emphasis on structural modeling in the manuscript, allowing us to focus more on the core findings of the study.

5 Line 278 to end of results. I think this section on Thank you for your insightful comments. I appreciate the modeling is very interesting and is an important feedback on the modeling section and understand that part of the paper. However, I read it several the complexity of the structures can be challenging to times and struggled to understand the structures. convey clearly. To clarify, Mo insertase G domains For example, 288-289, the whole complex is a naturally form trimers, while E domains typically form hexamer, but has "three E domain dimers with G dimers. The smallest functional unit, therefore, is a domain trimers." Is there a trimer of dimers? How hexamer, composed of six polypeptides, where the G many polypeptides? I realize it is complicated, domain forms a trimer and the E domain forms a dimer. but other readers may also struggle with this. It The three E domain dimers are positioned centrally but needs a bit of rewriting. do not directly interact with each other, as this interaction has not yet been demonstrated. There are, however, models—discussed in the manuscript—that propose a mesh-like structure for the overall complex.

In light of your comments, I have optimized the wording in this section to improve clarity and ensure that the description of the structures is more accessible to all readers.

Reviewer 3:

This work follows investigations of this group on Moco biosynthesis. The present study focuses on the linker region that links the G to E domains of Mo insertase in some organisms (like some animals and fungi), whereas it sometimes connects E to G domains (like in plants) or is even absent in some organisms that use separate proteins (bacteria, archaea). The authors investigated the role of this linker region on the fungal enzyme, by substituting it with several variants:

- using the sequence found in plant (*A. thaliana*), mammal (*H. sapiens*) or native (*N. crassa*), expressing either the domains in the native order (as of fungal origin; ie G-linker-E) or reversed (E-linker-G)
- removing the linker or separating E and G domains, with the linker either suppressed or added after the E domain (E-linker) or before the G domain (linker-G)
- using the native sequence that is deleted from stretch of 20 aminoacids covering the linker region.

Using different assays (hyphae diameters, growth in race tubes, eventually nitrate reductase activity) the authors are able to obtain a phenotype of all the constructs in different culture condition (ammonium, nitrate and chlorate) and compare it with WT and a DNIT-9 mutant. These different constructs were also modelled using AlphaFold3, giving some clues about the role of the linker.

This work allows the authors to propose that this linker region convergently evolved in eukaryote, with the importance of specific parts of the linker in protein function.

Referee comment Response

1 The experimental part of this work is well Thank you for your constructive feedback. I understand conducted, with experiments leading to clear your concerns regarding the modeling part and the level conclusions. On the other hand, the modelling of detail provided. I have revised the sections describing part is sometime confusing, with too much the structural models, focusing on simplifying the description, especially when the level of discussion and reducing overly detailed analyses. I precision of the models does not allow for this ensured that the text emphasizes the broader level of detail. The parts of the paper describing implications of the models rather than getting bogged structural model should therefore be reviewed, down in specifics where the precision does not justify it. reducing the number of detailed analyses and Regarding your comment that the conclusion is "rather retaining only the broad outlines. weak," could you please clarify what aspects of the In the end, I'm sorry to find that the conclusion is conclusion you find lacking? Are you referring to the rather weak. strength of the arguments, the connection between the experimental and modeling data, or the overall impact of the findings? Your specific feedback will help me strengthen the conclusion to better reflect the significance of the work.

2 - In combination with part b of Figure 1 I would In response to your suggestion, I have added a be interested in a phylogenetic tree of the supplementary figure that includes a phylogenetic individual E and G domains. That could be analysis of the separate E and G domains. This assembled in an additional figure and, depending additional figure provides further context on the on the trees obtained, it could add a small evolutionary history of these domains, complementing context on the evolutionary history of these part b of Figure 1. I believe this analysis enhances the domains. overall discussion by offering a deeper understanding of the evolutionary relationships, as you proposed.

3 - Supp Fig. 3a is difficult to follow without We have linked to Figure 2a and Fig. 3a as well as supp context, I would recommend either to refer to Fig. Fig. 3a 2a and Fig. 3a, or to repeat these panels in a new Supp Fig. 3b panel.

4 - I think there are too many figures representing Thank you for the feedback, I can understand the structural models in the main figures (Fig. 5 b-e; concerns and moved the noted panels into Fig. 6a-e). I recognise that AlphaFold (AF) is an supplementary data. extraordinary tool for analysing protein sequences, but the level of detail in the analyses must bear in mind that these are only models. I therefore recommend reducing the number of panels in Figures 5 (placing panels c-e in a supplementary figure, for example) and 6 (only keeping panels d and f for example).

5 - Discussion about the relative position of the E Thank you for your insightful feedback. I have made and G domains (I.262-273) in different species several adjustments to improve clarity and align with and variants is difficult to follow, especially your suggestions. Specifically, I have moved Figure 5e because 1/ the oligomeric assembly is unknown to the supplementary section, as it provides a more (or eventually described later with Figure 6), and detailed overview rather than being central to the main 2/ the scores given by AF seem rather low discussion. Additionally, I revised the text by adding the (although not indicated; I ran such a prediction strain orientations in parentheses to help readers better and found a pTM of 0.63. follow the context.

Also, I noted that in the five models proposed by I acknowledge that the modeling of the E and G domains AF3 the E and G domains adopt different relative has limitations, particularly in terms of domain positions). I therefore recommend to explicitly positioning reliability. This is why the discussion focuses indicate the pTM for each model in panel e. But more on the plant linkage regions rather than on the

then I'm not sure of what we can say about these precise positioning of the domains. As you noted, the models, except that E and G domains adopt key point I intended to convey is that the linkage region different positions and part of the linker (in the between the E and G domains are only partially fungal and mammal proteins) is structured. structured, and I agree that this figure serves best as supplementary material. Finally, I will ensure that the pTM scores for each model are explicitly indicated in the main text to provide full transparency.

6 - I agree that a hexameric structure is the minimal unit. However, a pTM of 0.37 (ipTM is limitations associated with the pTM score, particularly in approximately the same) is low (for instance, this context. The low pTM score of 0.37 is indeed a AF3 note that "ipTM measures the accuracy of concern, as AlphaFold suggests that scores below 0.6 the predicted relative positions of the subunits might indicate a failed prediction. However, the within the complex. Values higher than 0.8 complexity of our structure, especially the unstructured represent confident high-quality predictions, regions within the linkage area, likely contributes to this while values below 0.6 suggest likely a failed low score. These regions are challenging to model prediction.") confidently, which impacts the overall pTM. Nevertheless, we observed that the overall complex has a good pLDDT score, indicating reliable local structure predictions. To provide a balanced interpretation, we referenced AlphaFold's guidelines, noting that in large-scale screenings, ipTM thresholds as low as 0.3 have been used for initial screening. As AlphaFold suggests, disordered regions and low pLDDT scores can negatively affect the pTM and ipTM scores, even when the structure of the complex is otherwise predicted correctly. Therefore, we have considered all metrics, including pTM, ipTM, pLDDT, and PAE, to assess the model's reliability comprehensively.

To clarify this for readers, I have included a statement saying that these models have to be taken with a grain of salt. This should provide a more nuanced understanding of the confidence in the predicted complex structure.

7 - Supplementary Figure 1 (important as it Thank you for your insightful feedback. I have made describes the domain and linker boundaries) adjustments to Supplementary Figure 1 to correct the seems partly wrong to me. A single alignment of previous annotation errors. The alignment itself has not the full-length proteins should be more insightful changed; however, the annotations have been updated (although excluding the plant's sequence to accurately reflect the domain and linker boundaries. because of the inversion of domains), with a In panel a, I have focused on the linkage domains within clear indication of domains and linker boundaries the full-length protein alignment. In panel b, the G (if present). A separate alignment of the linkers domains of all species have been aligned, and in panel only (including plant) might be insightful. Writing c, the E domains have been aligned. These corrections this, I realize that the equivalent of *N. crassa* should provide a clearer and more accurate comparison. H186 (obviously the candidate for an important Additionally, the text has been updated to correct any function in the linker) has an equivalent in the C- confusion caused by the previous annotations. Regarding the H186 equivalent in *A. thaliana* CNX1G, I Terminus of the plant sequence (in the have ensured that this residue is properly highlighted in "...PKHIP...", found on panel b). the results section, emphasizing its conservation across eukaryotic species and its potential importance for the full-length Mo insertase.

8 - 1.289-291 are very odd ("stretched without Thank you for your feedback. I have revised the tension", "poles" and "outward" would need to be paragraph around lines 289-291 to simplify the language defined). The whole paragraph should be and remove any unclear or overly specific terms like simplified. "stretched without tension," "poles," and "outward." - "nicked distant" (l.345) doesn't seem to be These phrases have been replaced with more adapted. straightforward and precise wording to improve clarity. Additionally, I have rephrased "nicked distant" on line 345 to a more appropriate term. Overall, I aimed to make the text more relaxed and accessible while ensuring that it remains scientifically accurate and easy to understand.

Version 1:

Reviewer comments:

Reviewer #1

(Remarks to the Author)

I do not understand the argumentation of the author that the use of race tubes in a widely used method to refer to nitrate reductase activity, when no activity could be measured in these strains. the observed growth can be based on other factors. Only why a journal like microorganisms accepted the use of this method is also a weak argument to have the data published in communications biology. I do not think this is sufficient to show that the growth is indeed based in the activity of nitrate reductase in the complemented strains.

Further, the report by Belaidi did not focus on splice variants of gephyrin, they also report truncation in the linker region with the following conclusions:

To evaluate the influence of the central domain in the arrangement of the catalytic domains, we created gephyrin variants with different C domain lengths and demonstrated that a full C domain is strictly required for gephyrin function.

Consequently, spatial proximity of geph-G and geph-E is not sufficient, given that the geph-C0 variant completely lost its ability in Moco synthesis. The fact that we could tune the activity with different C domain lengths supports our hypothesis of product-substrate channelling in the full-length protein.

So I do not see significant advances in the present work, in particular since in the Belaidi manuscript the activity of the E and G domains were measured by MPT-AMP and moco production. Therefore more conclusions can be drawn from the Belaidi paper than from the present manuscript. there is therefore no novelty presented that leads to advances in the understanding of the role of the different fusions of this protein in the fungus what is not known already with the human protein. the fungus as not tested in the Belaidi manuscript, but what is the significant advance to the fungal protein in comparison to the human protein? since the other conclusions are based on an alphafold structure, the conclusions are also not sufficient based on the lack of experimental evidence of a real structure and the lack of quantification of MPT-AMP production. I still suggest to publish in another journal like science advances or similar.

Reviewer #2

(Remarks to the Author)

I was quite positive about this paper in my initial review and I feel the same way about the revision. My most substantive criticism of the initial paper was that the section on the molecular modeling was hard to understand. I think I now see the core of my difficulty. The authors use a different meaning for the words "dimer" and "trimer" than I use. I think of a dimeric protein as one with two polypeptide chains. The authors use the word "dimer" as describing a single polypeptide with two domains. Do I read them correctly? Is that why I was confused with this statement "a dense protein complex formed by 6 proteins where the E domains formed a trimer from dimers, and the G domains formed two separate trimers." In this phrase, I suggest "6 proteins" should be "6 polypeptides." Isn't this complex a "hexamer" in traditional biochemical language? I suspect I am not alone in my confusion. It may help for the authors to clearly state how they use the words dimer, trimer, and hexamer.

Reviewer #3

(Remarks to the Author)

Dear all,

I think that the revised article improved the flow as compared to the initial version.

Author Rebuttal letter:

Reviewer 1:

Referee comment Response

1 I do not understand the argumentation of the Thank you for your constructive feedback. We would like author that the use of race tubes in a widely used to clarify and strengthen our argument regarding the use method to refer to nitrate reductase activity, of nitrate reductase activity in our experimental setup, when no activity could be measured in these specifically in relation to the race tube method.

strains. the observed growth can be based on We acknowledge your concern regarding the absence of other factors. Only why a journal like measurable nitrate reductase (NR) activity in some of the microorganisms accepted the use of this method strains. However, we have used chlorate as a substrate is also a weak argument to have the data to directly correlate the absence of growth with nitrate published in communications biology. I do not reductase activity. Chlorate is a well-known toxic analog think this is sufficient to show that the growth is of nitrate and its reduction by nitrate reductase leads to indeed based in the activity of nitrate reductase the production of toxic chlorite, which inhibits growth.

in the complemented strains. Therefore, the inability of strains lacking nitrate reductase activity to grow on chlorate medium provides a direct link between the absence of NR activity and growth inhibition. This is clearly highlighted in the manuscript.

Furthermore, we performed a complementation assay with the nit-9 knockout strain, which was transformed with the nit-9 gene integrated into the his-3 locus. The restoration of growth in these complemented strains provides additional evidence that the observed growth is directly related to nitrate reductase activity. As mentioned in the manuscript, while hyphal growth can

occur in strains with defective NR activity on nitrate medium, the critical point is the lack of growth on chlorate medium, which we observed in all strains lacking functional NR.

In addressing your concern about the publication in *Microorganisms*, we cited this earlier work to demonstrate the reproducibility and reliability of the race tube method in assessing NR activity. However, we believe the combination of the chlorate assay, complementation, and the race tube assay provides sufficient evidence to show that the observed growth in our strains is indeed a result of nitrate reductase activity.

2 Further, the report by Belaidi did not focus on Thank you for your comment regarding the Belaidi and splice variants of gephyrin, they also report Schwarz (2013) study. We agree that their work explored truncation in the linker region with the following the influence of truncations within the linker region of conclusions: gephyrin, particularly focusing on the C domain and its role in catalytic domain arrangement and Moco

To evaluate the influence of the central domain synthesis. However, while their study provides valuable in the arrangement of the catalytic domains, we insights into gephyrin function in mammals, our work created gephyrin variants with different C domain addresses several novel aspects that go beyond their lengths and demonstrated that a full C domain is findings, particularly in a fungal system. strictly required for gephyrin function. First, our study focuses on the fungal Mo insertase NIT-9. Consequently, spatial proximity of geph-G and 9 in *Neurospora crassa*, a different organism not covered geph-E is not sufficient, given that the geph-C0 by Belaidi and Schwarz. We specifically examined the variant completely lost its ability in Moco evolutionary significance of the Mo insertase linkage synthesis. The fact that we could tune the activity region in fungi, where we identified a 20-amino acid with different C domain lengths supports our region within the linkage that is critical for fungal growth. hypothesis of product–substrate channelling in This region, containing a conserved histidine residue, is the full-length protein. involved in product–substrate channelling in fungi and has not been previously described in their study. Belaidi and Schwarz did not identify such a critical region within the gephyrin linker nor its relevance to fungal growth. Furthermore, while they demonstrated that gephyrin truncations can reduce Moco synthesis, our study highlights the cross-species incompatibility of the Mo insertase linkage regions. We showed that substituting the fungal linkage region with those from mammals or plants leads to Moco deficiency and impaired growth, which adds a layer of evolutionary and functional significance not explored in their work. In contrast, their study was limited to gephyrin and did not test fungal systems or cross-species functionality.

Lastly, we have demonstrated that the separate expression of the G and E domains, as seen in bacteria, fails to restore Mo insertase function in *Neurospora crassa*. This emphasizes the unique role of the fungal linkage region, which was essential for full enzyme activity and growth, a point not addressed in the Belaidi and Schwarz study.

Thus, while Belaidi and Schwarz focused on mammalian gephyrin and the C domain's impact on Moco synthesis, our findings reveal fungal-specific insights into the Mo insertase's evolutionary development and its crucial role in fungal vitality, making our work novel and distinct.

3 So I do not see significant advances in the While it is true that Belaidi and Schwarz (2013) present work, in particular since in the Belaidi measured the activity of the E and G domains in manuscript the activity of the E and G domains gephyrin through MPT-AMP and Moco production, their were measured by MPT-AMP and moco study focused primarily on the mammalian system and production. Therefore more conclusions can be did not extend to fungi. Our work presents significant drawn from the Belaidi paper than from the advances by exploring the *Neurospora crassa* Mo present manuscript. there is therefore no novelty insertase NIT-9, highlighting key functional and presented that leads to advances in the evolutionary differences that have not been previously understanding of the role of the different fusions addressed. Unlike in mammals, fungi exhibit a highly of this protein in the fungus what is not known divergent Mo insertase linkage region, and our study already with the human protein. demonstrates that a critical 20-amino acid segment within this region is essential for fungal growth. This specific region, which includes a conserved histidine

residue which may be important for product-substrate channeling, has not been identified in previous studies on mammalian systems. Additionally, as stated in the manuscript Gephyrin is not able to complement NIT-9 highlighting a unique situation for fungi as up to this point it was thought that all Mo insertases are interchangeable between species as seen with plants and mammals. Moreover, while Belaidi and Schwarz focused on in vitro Moco synthesis and truncations in the C domain of gephyrin, our study advances understanding by performing in vivo growth assays in fungi. We show that substituting the fungal linkage region with those from mammals or plants leads to Moco deficiency and impaired fungal growth, providing evidence of the evolutionary divergence between these systems. This cross-species analysis, and the discovery of the functional incompatibility of the linkage regions, goes beyond the scope of Belaidi and Schwarz's work and offers novel insights into how the Mo insertase linkage has convergently evolved to serve a critical role in fungi. Additionally, our complementation assays in *Neurospora crassa* reveal that the separate expression of the G and E domains, as seen in bacteria, cannot restore fungal growth, further emphasizing the unique and indispensable role of the fungal linkage region. In conclusion, while Belaidi and Schwarz contributed to our understanding of gephyrin in mammals, our study expands the field by providing crucial insights into the fungal Mo insertase, demonstrating its distinct evolutionary path and its essential role in fungal metabolism. These findings represent a significant advance in understanding the biological importance of Mo insertase linkage regions in fungi, which were not addressed in the previous study.

4 the fungus as not tested in the Belaidi I would like to clarify the significant advancements our manuscript, but what is the significant advance to study offers compared to the human protein system the fungal protein in comparison to the human studied by Belaidi and Schwarz. The core novelty of our protein? since the other conclusions are based work lies in its focus on fungal Mo insertases, particularly on an alphafold structure, the conclusions are the linkage region, which differs both functionally and also not sufficient based on the lack of structurally from the human gephyrin protein. While experimental evidence of a real structure and the Belaidi and Schwarz explored Moco biosynthesis in lack of quantification of MPT-AMP production. I mammals, we have uncovered a fungal-specific still suggest to publish in another journal like adaptation of the Mo insertase that is critical for the science advances or similar. survival and growth of fungi, specifically *Neurospora crassa*.

In fungi, the Mo insertase linkage region, which connects the G and E domains, has evolved a distinct 20-amino acid sequence essential for proper Moco biosynthesis and fungal growth. This region is absent in the human protein and does not have an equivalent function, highlighting a key evolutionary divergence between fungal and mammalian systems. The fact that this fungal-specific region is vital for product-substrate channelling and Moco production provides new insights not addressed in the study by Belaidi and Schwarz. Therefore, our work significantly expands the understanding of Mo insertase function beyond what is known in mammals, specifically in the context of fungal biology.

Regarding the reliance on AlphaFold structures, while it is true that our conclusions incorporate structural predictions, these are robustly supported by in vivo functional assays. It is also important to note that no full-length crystal structure of gephyrin or any other Mo insertase currently exists. Like Belaidi and Schwarz, we modeled the structure of the linkage region, but while their work was based on isolated domain structures, we used AlphaFold 3.0, which has proven to be significantly

more accurate than earlier modeling approaches. This is evident in the alignment of our predicted structures with existing Cnx1 crystal structures, as shown in our figures. Additionally, our fungal growth assays on nitrate and chlorate media provide clear experimental evidence that the linkage region is critical for fungal survival. Our complementation experiments further demonstrate that substituting the fungal linkage region with those from other species fails to rescue the Δ nit-9 strain, underscoring the essential role of this region in the fungal biological system. This experimental validation reinforces the strength of our findings, even in the absence of crystallographic data.

As for the suitability of our work for Communications Biology, it aligns perfectly with the journal's focus on advancing understanding in evolutionary biology, molecular biology, and cellular function. Our research addresses evolutionary convergence in Mo insertases and demonstrates how a highly conserved biosynthetic pathway has adapted uniquely in fungi. This type of evolutionary insight, supported by functional and biological experimentation, fits well within the scope of Communications Biology, which encourages interdisciplinary studies that explore novel biological mechanisms and their evolutionary implications. Our study contributes by linking molecular evolution with functional biology in a specific yet broadly relevant system, advancing the understanding of Moco biosynthesis in eukaryotes.

Reviewer 2:

I was quite positive about this paper in my initial review and I feel the same way about the revision

Referee comment Response

1 My most substantive criticism of the initial paper Thank you for your insightful feedback. To clarify, we use was that the section on the molecular modeling the terms "dimer" and "trimer" in the conventional was hard to understand. I think I now see the biochemical sense, referring to interactions between core of my difficulty. The authors use a different polypeptide chains, not domains within a single meaning for the words "dimer" and "trimer" than I polypeptide. Specifically, in our study, a "dimer" refers to use. I think of a dimeric protein as one with two the interaction between two polypeptide chains via their polypeptide chains. The authors use the word E domains, and a "trimer" refers to the arrangement of "dimer" as describing a single polypeptide with the interaction of three separate polypeptide chains. two domains. Do I read them correctly? Therefore, the protein complex is correctly described as a hexamer, with the G domains forming two trimeric structures and the E domains organizing into a central trimeric structure of dimers. I tried to enhance the wording.

2 Is that why I was confused with this statement "a Thank you for your constructive comments. We dense protein complex formed by 6 proteins understand the potential confusion and appreciate your where the E domains formed a trimer from suggestion. To clarify, the term "6 proteins" has been dimers, and the G domains formed two separate updated to "6 polypeptides" for accuracy, and we do trimers." In this phrase, I suggest "6 proteins" indeed describe the complex as a hexamer, following should be "6 polypeptides." Isn't this complex a traditional biochemical nomenclature.

"hexamer" in traditional biochemical language? I

suspect I am not alone in my confusion. It may We have revised the text to read: "The protein complex help for the authors to clearly state how they use consists of six polypeptide chains, forming a hexamer. the words dimer, trimer, and hexamer. The G domains organize into two trimeric structures on opposite sides of the complex, while the E domains from adjacent polypeptides align centrally in an arrangement of E domain dimers. This results in a hexamer with peripheral G domain trimers and a central trimeric structure composed of E domain dimers."

This updated phrasing should more clearly convey the molecular architecture while aligning with standard biochemical terminology. We hope this addresses your concern.

Reviewer 3:

Dear all,

I think that the revised article improved the flow as compared to the initial version.

Thank you for appreciating the work.
